# Calving from horizontal forces in a revised crevasse-depth framework

Donald A. Slater[1] and Till J. W. Wagner[2]

[1]School of Geosciences, University of Edinburgh, Edinburgh, UK
[2]Department of Atmospheric and Oceanic Sciences, University of Wisconsin–Madison, Madison, USA

**Correspondence:** Donald A. Slater (donald.slater@ed.ac.uk)

**Abstract.** Calving is a key process for the future of our ice sheets and oceans, but representing it in models remains challenging. Among numerous possible calving parameterisations, the crevasse-depth law remains attractive for its clear physical interpretation and its performance in models. In its classic form, however, it requires ad-hoc and arguably unphysical modifications to produce crevasses that are deep enough to result in calving. Here, we adopt a recent analytical approach accounting for the feedback between crevassing and the stress field at floating ice shelves, and generalise the framework by adding non-zero ice tensile strength and basal friction and applying it to grounded marine-terminating glaciers. We show that the revised formulation removes the need for ad-hoc modifications and predicts that full-thickness calving should occur when grounded glaciers reach flotation provided the calving front ice thickness is greater than around 400 m. The revised formulation also predicts no calving for ice thinner than around 400 m, suggesting that calving at such glacier fronts is not driven purely by horizontal forces. We find good observational support from both grounded marine-terminating glaciers and floating ice shelves for this analysis. We advance the revised crevasse-depth formulation as a step towards understanding differing calving styles and a better representation of calving in numerical models.

## 1 Introduction

Glacier and ice shelf calving plays an important role in our climate system. This is most prominently due to its influence on mass loss from bodies of ice and thereby sea-level rise (Fox-Kemper, 2021), but also due to the climate and ecosystem effects of iceberg melting over the polar ocean (see review by Alley et al., 2023). Beyond its significance in the physical world, calving takes on an outsized role in climate modelling as it presents a crucial boundary condition linking ice sheet and ocean models (Benn et al., 2017; Alley et al., 2023).

Calving is a complex phenomenon that varies in character in both space and time (e.g., Benn et al., 2007; Bassis and Jacobs, 2013) and is influenced by surface melting (Cook et al., 2012), wave action (Petlicki et al., 2015), tides (Holmes et al., 2023), submarine melting (Luckman et al., 2015), buttressing by mélange and sea ice (Miles et al., 2017; Wehrlé et al., 2023), ice properties (e.g., Borstad et al., 2017) and the dynamics of rifts (Larour et al., 2021). In Greenland, most calving occurs at the front of grounded tidewater glaciers. It can be characterised as consisting of either (i) low volume but high frequency events, sometimes generally called serac failure (e.g., How et al., 2019); or (ii) high volume but low frequency, full-thickness events

(e.g., James et al., 2014), though there are other ways of characterising calving styles and a continuum of possibilities may be more realistic (e.g., Alley et al., 2023). Full-thickness calving tends to be more common for glacier termini that are close to or at flotation, and in particular is the style of calving that leads to the loss of tabular icebergs from Antarctica's ice shelves. At the level of the stresses experienced by the ice, calving can be driven by horizontal (van der Veen, 1998; Todd et al., 2018), vertical (Ma et al., 2017; Slater et al., 2021) and rotational forces (Wagner et al., 2016; Sartore et al., 2025), or perhaps by any combination of the three (Bassis and Walker, 2012; Ma and Bassis, 2019; Cowton et al., 2019; Schlemm and Levermann, 2019).

The search for better fundamental understanding of the calving process has seen the use of damage and linear elastic fracture mechanics approaches to investigate the nucleation and propagation of ice cracks (Duddu et al., 2013; Albrecht and Levermann, 2014; Yu et al., 2017; Lai et al., 2020; Zarrinderakht et al., 2022; Gao et al., 2023). Complementary to this, modeling approaches that treat glaciers as a large number of discrete ice "particles" allow a range of calving behaviours to emerge as a result of the modeled stresses and the breaking of bonds between the particles (Bassis and Jacobs, 2013; Åström et al., 2014; van Dongen et al., 2020b). High-resolution, three-dimensional continuum modeling gives detailed insight into stresses within the ice and – after adoption of a calving criterion – suggests how calving may respond to geometry and environmental forcings (e.g., Todd et al., 2018; Holmes et al., 2023).

Without further parameterisation of their results, however, such approaches are not realistically implementable as calving conditions in large-scale ice sheet models, such as the continent-scale simulations run as part of the Ice Sheet Model Inter-comparison Project (ISMIP6; Goelzer et al., 2020; Seroussi et al., 2020). The challenge in formulating such calving conditions is to find a unifying criterion that is general enough to capture the dominant calving behavior for a wide variety of physical settings yet simple enough to be represented in continuum models that may struggle for resolution at the marine boundary and may not solve for all components of the stress tensor. In particular, to render long-term continent-scale simulations computationally feasible, most models are simplified to a depth-integrated framework. As a result, a calving law that draws solely on depth-integrated quantities is highly sought after.

Many such depth-integrated calving laws have been proposed, often grouped into "rate" and "position" laws (Amaral et al., 2020), though there is a close relationship between these groups. Rate laws conceptualise calving as a process that removes mass from the calving front at a given rate and thus characteristically consider strain rates and flow velocities (e.g., Levermann et al., 2012; Morlighem et al., 2016; Mercenier et al., 2018). Position laws conceptualise calving as a process that removes ice to a given position upstream from the terminus and characteristically consider geometry and stress thresholds (e.g., Van der Veen, 1996, 2002; Pfeffer et al., 1997; Benn et al., 2007). The approach used in this study – the relation between modeled crevasse penetration depths and calving – is an example of a position law.

Nye (1955) laid the foundation of how stress balances in the ice sheet could be applied to estimate maximum stable surface crevasse depths. Weertman (1973) added hydrofracture – the deepening of surface crevasses due to the presence of meltwater – while Jezek (1984) applied a similar framework to basal crevasses. Benn et al. (2007) linked crevasse depths to the calving process, suggesting that calving could be assumed to occur when the surface crevasses reached the sea surface waterline (rather than requiring full-depth penetration). This concept was developed further by Nick et al. (2010), who also accounted for the

potential presence of basal crevasses and took calving to occur when surface and basal crevasses meet. Versions of the law developed by Nick et al. (2010) have been widely used since (e.g., Nick et al., 2013; Todd et al., 2019; Cook et al., 2022; Holmes et al., 2023).

Rate and position calving laws have been compared and validated against observations. On the one hand, Choi et al. (2018) and Wilner et al. (2023) conclude that rate calving laws result in the closest matches between modelled and observed calving

front positions for specific Greenlandic glaciers and Antarctic ice shelves. On the other hand, based purely on observations, Amaral et al. (2020) argue that the crevasse-depth law of Nick et al. (2010) is generally highly accurate. They find this law to be relatively insensitive to imperfect parameter calibration, so that a single parameter value can readily be applied to many glaciers, while rate laws require more glacier-specific tuning. The crevasse-depth approach thus remains a leading contender for a high-fidelity, low-complexity calving parameterisation.

Despite this conclusion, the crevasse-depth law remains hamstrung by the fact that in its basic formulation, the modelled crevasse depths close to the calving front are not deep enough to drive calving. This has led to understandable but unsatisfactory modifications, such as the proposal that calving occurs when surface crevasses reach the sea surface (Benn et al., 2007), without detailed physical consideration of how the remaining ice fractures, or the use of meltwater as a tuning parameter to deepen surface crevasses, with little observational evidence linking the presence of meltwater in surface crevasses to calving (Amaral

et al., 2020; Enderlin and Bartholomaus, 2020).

Here, we revisit the original crevasse-depth law, but account for the feedback between crevasses and the stress field, and for variable density of water in basal crevasses, with an approach recently employed by Buck (2023) and Coffey et al. (2024) in the context of ice shelf rifting. We adapt the approach to grounded tidewater glaciers and extend it by considering ice tensile strength and basal friction. We find that these modifications generally lead to significantly larger crevasses than the original

law and can give full-depth crevassing without appealing to water in surface crevasses. Furthermore, the resulting "modified" crevasse-depth law has interesting properties that show promise at explaining differing calving styles. This paper proceeds by first presenting the original crevasse-depth law, then the modification, then the results: predicted crevasse sizes, the resulting calving criterion and the sensitivity to physical parameters. We discuss observational support for the findings and conclude by placing our study in the context of the ongoing search for better calving parameterisations.

## 2 Methods

### 2.1 Set-up

We consider a grounded flow-line marine-terminating glacier with calving front at $x = x_{cf}$, sea level at $z = 0$ and constant ice density $\rho_i$ (Fig. 1). The bed topography in the marine-based part of the glacier is $z = -w(x)$ so that the ocean depth at the calving front is $w(x_{cf})$. The ice thickness is $H(x)$, giving an ice surface elevation above sea level $h(x) = -w(x) + H(x)$. It

is assumed there is an open hydraulic connection in terms of pressure from the bed below the glacier to the calving front, so that where the bed is below sea level, the water pressure at the base of the glacier is $\rho_w g w(x)$, with density of seawater $\rho_w$ and gravitational acceleration $g$. The depth of surface crevasses is denoted $d_s(x)$ and the height of basal crevasses is $d_b(x)$;

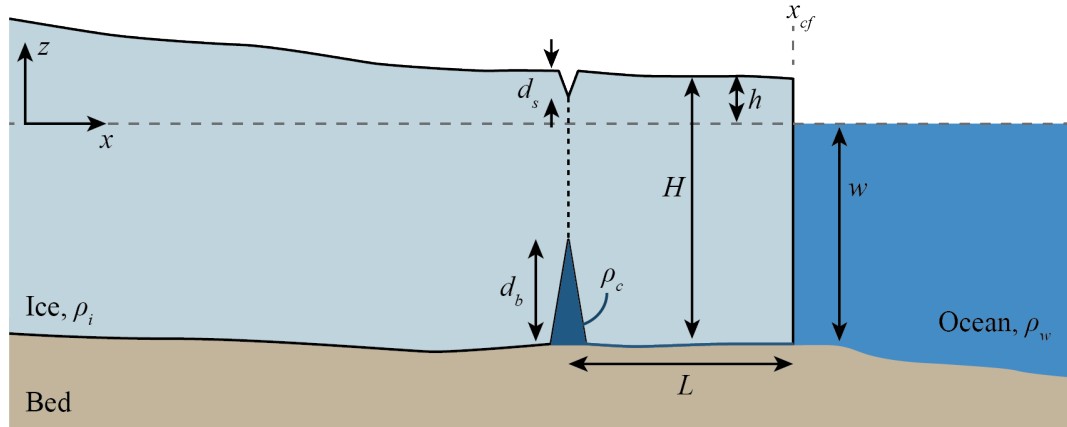

**Figure 1.** Schematic of a grounded marine-terminating glacier, having thickness $H(x)$ and density $\rho_i$. Indicated are the ocean water depth $w(x)$, height of ice surface above sea level $h(x)$, surface crevasse penetration depth $d_s$, and basal crevasse height $d_b$. Ocean water has density $\rho_w$ and water in the basal crevasse has density $\rho_c$. The horizontal distance between the crevasse pair and the calving front $x_{cf}$ is $L$, and over this distance the bed exerts a drag $\tau_b$ on the ice.

when referring to both surface and basal crevasses we use 'size' rather than 'depth' or 'height'. Surface crevasses are assumed to be dry and basal crevasses are assumed to be filled with water of density $\rho_c$. For the numerical results in this paper, we take $g = 9.81\ \mathrm{m\,s^{-2}}$, $\rho_i = 917\ \mathrm{kg\,m^{-3}}$ and $\rho_w = 1027\ \mathrm{kg\,m^{-3}}$ throughout; all other parameters are varied.

Under the hydrostatic approximation, the vertical normal stress in the glacier is given by

$$\sigma_{zz}(x,z) = -\rho_i g[h(x) - z]. \tag{1}$$

Splitting the stress into deviatoric and pressure parts ($\sigma_{zz} = \tau_{zz} - p$ and $\sigma_{xx} = \tau_{xx} - p$), and taking ice to be incompressible ($\tau_{xx} + \tau_{zz} = 0$), the horizontal normal stress is then (see, e.g., Greve and Blatter, 2009)

$$\sigma_{xx}(x,z) = \tau_{xx} - p = \tau_{xx} - \tau_{zz} + \sigma_{zz} = 2\tau_{xx} - \rho_i g[h(x) - z] \equiv R_{xx} - \rho_i g[h(x) - z], \tag{2}$$

where in the last equality we have introduced the resistive stress, $R_{xx} = 2\tau_{xx}$, that is commonly used in frameworks for calving. Part of our motivation is to further the implementation of calving laws in the depth-integrated models that are used for large-scale simulation of glaciers and ice sheets. Such models solve only for the depth-integrated value of $\tau_{xx}$, hence we assume that $R_{xx}$ does not vary with $z$ but return to this point in the discussion.

Before applying the revised crevasse-depth framework to grounded glaciers, we recap the classic crevasse-depth calving law which provides the starting point for the modification. We use the terminology 'classic' because this form, or minor variations on it, have been widely adopted (e.g. Nye, 1955; Weertman, 1973; Nick et al., 2010).

## 2.2 Classic crevasse-depth calving law

Within the classic crevasse-depth calving law, dry surface crevasses are assumed to penetrate to a depth where the horizontal normal stress $\sigma_{xx}$ vanishes. Using Eq. 2, the fractional surface crevasse depth is then (Nye, 1955)

$$\frac{d_s}{H} = \frac{R_{xx}}{\rho_i g H}. \tag{3}$$

Basal crevasses are assumed to penetrate to a height above the bed where the horizontal normal stress plus the crevasse water pressure vanishes:

$$R_{xx} - \rho_i g (H - d_b) + \rho_w g w - \rho_c g d_b = 0. \tag{4}$$

Solving Eq. 4 gives

$$\frac{d_b}{H} = \frac{\rho_i}{\rho_c - \rho_i} \left( \frac{R_{xx}}{\rho_i g H} - \frac{H_{ab}}{H} \right), \tag{5}$$

unless $R_{xx}/(\rho_i g H) < H_{ab}/H$, in which case there are no basal crevasses. The height above buoyancy, $H_{ab}$, is given by

$$H_{ab} = \begin{cases} 0 & \text{when } H \leq \frac{\rho_w}{\rho_i} w \text{ (ice is floating)}, \\ H - \frac{\rho_w}{\rho_i} w & \text{when } H > \frac{\rho_w}{\rho_i} w \text{ (ice is grounded below sea level)}. \end{cases} \tag{6}$$

Note that in the case of a basal crevasse filled by seawater, where $\rho_c = \rho_w$, Eq. 4 reduces to the basal crevasse heights of Weertman (1973) and later Nick et al. (2010). In what follows, we will wish to allow for a variable density of water in basal crevasses, hence we retain $\rho_c$ as an independent parameter that may differ from $\rho_w$, but we will still refer to Eqs. 3 & 5 as the classic crevasse-depth calving law. The total crevassed fraction, $f = (d_s + d_b)/H$, is the sum of Eqs. 3 & 5 and is given by

$$f = \begin{cases} \frac{R_{xx}}{\rho_i g H} & \text{when } \frac{R_{xx}}{\rho_i g H} \leq \frac{H_{ab}}{H}, \\ \frac{\rho_c}{\rho_c - \rho_i} \frac{R_{xx}}{\rho_i g H} - \frac{\rho_i}{\rho_c - \rho_i} \frac{H_{ab}}{H} & \text{when } \frac{H_{ab}}{H} < \frac{R_{xx}}{\rho_i g H} < 1 + \frac{\rho_i}{\rho_c} \left( \frac{H_{ab}}{H} - 1 \right), \\ 1 & \text{when } \frac{R_{xx}}{\rho_i g H} \geq 1 + \frac{\rho_i}{\rho_c} \left( \frac{H_{ab}}{H} - 1 \right), \end{cases} \tag{7}$$

where the first case corresponds to surface crevasses but no basal crevasses, the second case to surface and basal crevasses, and the third case to full crevasse penetration and calving.

## 2.3 Resistive stress close to the calving front

In general, the resistive stress that determines crevasse depths is obtained using a numerical ice flow model. However, we can estimate the resistive stress close to the calving front without a numerical model using the boundary condition at the calving front that $\sigma_{xx}(x_{cf}, z) = 0$ above the water and $\sigma_{xx}(x_{cf}, z) = \rho_w g z$ below the ocean surface. If, between the calving front and the crevasses, the ice-bed drag is $\tau_b$ (Fig. 1), then the resistive stress at the crevasses can be estimated as

$$R_{xx}(x_{cf} - L) \approx \begin{cases} \frac{1}{2} \rho_i g H \left( 1 - \frac{\rho_i}{\rho_w} \right) & \text{when } H \leq \frac{\rho_w}{\rho_i} w \text{ (ice is floating)} \\ \frac{1}{2} \rho_i g H \left( 1 - \frac{\rho_w}{\rho_i} \frac{w^2}{H^2} \right) - \frac{L}{H} \tau_b & \text{when } H > \frac{\rho_w}{\rho_i} w \text{ (ice is grounded)} \end{cases} \tag{8}$$

where $w$ and $H$ take their values at the calving front and $L$ is the horizontal distance between the crevasses and the calving front. Eq. 8 is used throughout this manuscript to estimate crevasse sizes close to the front of marine-terminating glaciers.

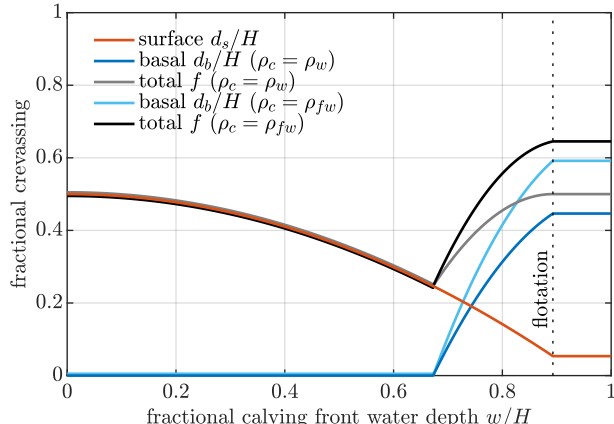

**Figure 2.** Dependence of crevasse sizes on fractional calving front water depth in the classic crevasse-depth law. This applies only close to the calving front and assumes no basal friction. A fractional calving front water depth $w/H = 0$ corresponds to a dry calving cliff, while $w/H = 0.5$ corresponds to a calving front where the ocean depth is half of the ice thickness and a water depth $w/H > \rho_i/\rho_w \approx 0.89$ corresponds to a floating calving front. When the basal crevasse is filled with seawater we get the basal crevassing indicated in dark blue and total crevassing indicated in grey. When the basal crevasse is filled with freshwater we get the basal crevassing indicated in light blue and total crevassing indicated in black. Surface crevasse depths (red) are independent of basal crevasse water density. Note that this plot applies regardless of ice thickness.

Crevasse depths in the classic crevasse-depth law are obtained by substituting Eq. 8 into Eqs. 3, 5 & 7. The resulting crevasse
sizes are shown in Fig. 2 as a function of fractional calving front water depth $w/H$, for the two end-member cases where the basal crevasse is filled with seawater ($\rho_c = \rho_w$) or with freshwater ($\rho_c = \rho_{fw} = 1000$ kg m$^{-3}$), and we have neglected basal friction ($\tau_b = 0$). Surface crevassing is independent of basal crevasse water density, and is greatest at a dry calving cliff ($w/H = 0$) when there is no ocean water to support the cliff and the resistive stress is largest. As fractional calving front water depth increases, fractional surface crevasse depth decreases, until for glaciers at flotation ($w/H = \rho_i/\rho_w \approx 0.89$) the fractional
surface crevassing is roughly 0.06. At dry or well-grounded calving fronts the water depth is insufficient to give basal crevasses; these arise when fractional water depths exceed $w/H \approx 0.66$. Above this threshold, basal crevasses grow quickly with water depth until at flotation they reach a fractional height of roughly 0.44 when filled with seawater, or roughly 0.58 when filled with freshwater.

A key point from Fig. 2 for the present study is that the total fractional crevassing never reaches 1. For the case of seawater-
filled basal crevasses, the maximum crevassed fraction is $f = 0.5$, occurring at a dry front entirely due to surface crevassing, or at a floating front due mostly to basal crevassing. If basal crevasses are filled with freshwater, then the water pressure in the basal crevasse falls away more slowly with height, giving larger basal crevasses. However, the total fractional crevassing still

does not exceed $f \approx 0.66$, occurring for glaciers at flotation (Fig. 2). The inclusion of non-zero basal friction would decrease the resistive stress and lead to even smaller crevasses.

In its classic form, therefore, the crevasse-depth calving law applied to a calving front predicts that calving should never occur. This has motivated the modifications described in the introduction, which either add more stress to the formulation (in the case of water in surface crevasses, Weertman, 1973; Nick et al., 2010) or relax the requirement for calving to occur only when $f = 1$ (in the case of calving once a surface crevasse reaches the waterline, Benn et al., 2007).

An alternative approach, proposed by Buck (2023) for the case of rifts at floating ice shelves, is to account for the concen-
tration of horizontal stress due to the presence of crevassing, and the feedback of this concentration on crevasse size. We now apply that approach to grounded tidewater glaciers.

## 2.4    Modified formulation for crevasse sizes

Following Buck (2023), we assume that the presence of crevasses modifies the horizontal normal stress to $\sigma'_{xx}$. If hydrostatic balance still applies, the vertical normal stress is unmodified and following Eq. 2, we assume that the modified horizontal
normal stress in the intact ice – between the surface and basal crevasses – can be written

$$\sigma'_{xx}(x,z) = R'_{xx}(x) - \rho_i g[h(x) - z], \tag{9}$$

where $R'_{xx}$ is the modified resistive stress in the intact ice. In the crevassed ice, the horizontal normal stress has to balance the fluid pressure in the crevasses, giving in total

$$\sigma'_{xx}(x,z) = \begin{cases} 0 & h \geq z > h - d'_s, \\ R'_{xx}(x) - \rho_i g[h(x) - z] & h - d'_s > z > -w + d'_b, \\ \rho_c gz - (\rho_w - \rho_c)gw & -w + d'_b \geq z > -w, \end{cases} \tag{10}$$

where $d'_s$ and $d'_b$ are the modified surface and basal crevasse sizes. This contrasts with the classic law, in which $\sigma_{xx}(x,z) = R_{xx}(x) - \rho_i g[h(x) - z]$ at all depths (Eq. 2). Continuing to follow Buck (2023), we insist that the horizontal force balance is conserved under crevassing, so that

$$\int_{-w(x)}^{h(x)} \sigma_{xx}(x,z) \, dz = \int_{-w(x)}^{h(x)} \sigma'_{xx}(x,z) \, dz. \tag{11}$$

Performing the integrals leads to the horizontal force balance

$$\left(H - d'_s - d'_b\right) R'_{xx} = H R_{xx} - \frac{1}{2}\rho_i g d'^2_s - \frac{1}{2}\rho_i g \left(2Hd'_b - d'^2_b\right) + \rho_w g w d'_b - \frac{1}{2}\rho_c g d'^2_b. \tag{12}$$

The modified crevasse sizes themselves are then defined just as for the classic law (Eqs. 3 & 5) but replacing the resistive stress with the modified resistive stress. For reasons that will become apparent later, we also include a tensile strength for ice, $\sigma_{max}$, so that the modified surface crevasse depth is

$$\frac{d'_s}{H} = \frac{R'_{xx} - \sigma_{max}}{\rho_i g H}, \tag{13}$$

and the modified basal crevasse height is

$$\frac{d'_b}{H} = \frac{\rho_i}{\rho_c - \rho_i} \left( \frac{R'_{xx} - \sigma_{max}}{\rho_i g H} - \frac{H_{ab}}{H} \right). \tag{14}$$

Eqs. 12, 13 & 14 are then three equations in the three unknowns $R'_{xx}$, $d'_s$ and $d'_b$, which can be solved analytically to give the modified crevasse sizes. Defining for convenience $\tilde{R}_{xx} = R_{xx}/(\rho_i g H)$ and $\tilde{\sigma}_{max} = \sigma_{max}/(\rho_i g H)$, the solution has four cases:

1. No crevasses are present when $R_{xx} \le \sigma_{max}$. That is, when the unmodified resistive stress is smaller than the ice tensile strength. In this case, the crevasse sizes are

$$d'_s = d'_b = 0. \tag{15}$$

2. Surface crevasses are present, but basal crevasses are not, when

$$0 < \tilde{R}_{xx} - \tilde{\sigma}_{max} \le \frac{H_{ab}}{H} \left( 1 - \tilde{\sigma}_{max} - \frac{1}{2} \frac{H_{ab}}{H} \right), \tag{16}$$

that is, when the resistive stress is large enough to give surface crevasses but the ice is grounded in sufficiently shallow water to suppress basal crevasses. In this case, the basal crevasse height is $d'_b = 0$ and the surface crevasse depth is given by

$$\frac{d'_s}{H} = 1 - \tilde{\sigma}_{max} - \sqrt{1 - 2\tilde{R}_{xx} + \tilde{\sigma}^2_{max}}. \tag{17}$$

3. Both surface and basal crevasses are present when

$$\tilde{R}_{xx} - \tilde{\sigma}_{max} > \frac{H_{ab}}{H} \left( 1 - \tilde{\sigma}_{max} - \frac{1}{2} \frac{H_{ab}}{H} \right), \tag{18}$$

that is, when the ice is sufficiently close to flotation to allow basal crevasses. In this case, the basal crevasse height is given by

$$\frac{d'_b}{H} = \frac{\rho_i}{\rho_c} \left[ 1 - \frac{H_{ab}}{H} - \frac{\rho_c \tilde{\sigma}_{max}}{\rho_c - \rho_i} - \sqrt{1 + \frac{2\rho_i}{\rho_c - \rho_i} \frac{H_{ab}}{H} \left( 1 - \frac{1}{2} \frac{H_{ab}}{H} \right) - \frac{2\rho_c \tilde{R}_{xx}}{\rho_c - \rho_i} + \left( \frac{\rho_c \tilde{\sigma}_{max}}{\rho_c - \rho_i} \right)^2} \right], \tag{19}$$

from which the surface crevasse depth can be obtained using

$$\frac{d'_s}{H} = \frac{\rho_c - \rho_i}{\rho_i} \frac{d'_b}{H} + \frac{H_{ab}}{H}. \tag{20}$$

4. Finally, there is no solution when

$$\tilde{R}_{xx} > \frac{\rho_c - \rho_i}{2\rho_c} + \frac{\rho_i}{\rho_c} \frac{H_{ab}}{H} \left( 1 - \frac{1}{2} \frac{H_{ab}}{H} \right) + \frac{\rho_c \tilde{\sigma}^2_{max}}{2(\rho_c - \rho_i)}, \tag{21}$$

that is, when the term in the square root in Eq. 19 becomes negative. In this case, the resistive stress is sufficiently large that there is no configuration of crevasse sizes that can satisfy the horizontal force balance in Eq. 12.

The potential for the crevasse sizes to be undefined, as in solution case 4, is a behaviour that does not appear in the classic crevasse depth law. This lack of solution indicates that the horizontal forces on the nascent calving block (Fig. 1) cannot be balanced. If we included an inertial term in the force balance, we would find that the block accelerates away from the rest of the glacier. Such a situation has been observed as an immediate precursor to calving (e.g., van Dongen et al., 2020a) and therefore we interpret this lack of solution and the undefined crevasse sizes as calving and explore this further in section 3.3.

As before, we note that in general, an ice flow model would provide the resistive stress $R_{xx}$ that is required to estimate the crevasse depths given by Eqs. 15-21. But again, we can proceed without an ice sheet model and see the implications of the modified calving law by using the expression in Eq. 8 for the resistive stress close to the calving front. The crevasse depths close to the calving front are then principally a function of the frontal water depth $w$ (which controls both the resistive stress $R_{xx}$ and the height above buoyancy $H_{ab}$), modulated by the chosen values for parameters including the water density in the basal crevasse $\rho_c$, the ice tensile strength $\sigma_{max}$, and the basal friction $\tau_b$.

At this point, we highlight differences in our framework relative to the floating ice shelf case of Buck (2023) and Coffey et al. (2024): those studies consider ice that is at flotation and assume zero tensile strength, so the frontal water depth $w$, basal friction $\tau_b$, and ice tensile strength $\sigma_{max}$ do not appear in their analysis. Furthermore, those studies treat the resistive stress $R_{xx}$ as an independent parameter to be varied. Here, consistent with our desire to investigate calving across a range of grounded glaciers, we adopt the frontal water depth $w$ as our principal independent parameter, and the resistive stress follows as a dependent parameter following Eq. 8. This has two important implications: first, the resistive stresses considered here are consistently higher than those in the ice shelf studies; and second, in our study the resistive stress and the basal crevasse water pressure are linked as they both depend on the frontal water depth, whereas in the ice shelf studies the basal crevasse water pressure is independent of the resistive stress. These differences in framework arise naturally when moving from floating to grounded ice, and the connection of our expressions to the floating cases of Buck (2023) and Coffey et al. (2024) is given in Appendix A.

The remainder of this paper considers the implications of applying the modified crevasse-depth framework to the calving fronts of grounded marine-terminating glaciers (with flotation as a limiting case). All results are obtained by substituting Eqs. 6 & 8 into Eqs. 15-21.

## 3 Results

The results section proceeds by first showing an illustrative example of the modified crevasse sizes (section 3.1) and then exploring the parameter sensitivity of the modified crevasse sizes (section 3.2). After building this understanding of the modified crevasse sizes we then consider the implications of this modified framework for calving (section 3.3).

### 3.1 An illustrative example

The modified crevasse sizes are a complex function of calving front water depth $w$, ice thickness $H$, tensile strength $\sigma_{max}$, crevasse water density $\rho_c$, basal friction $\tau_b$, and crevasse spacing $L$. This contrasts with the classic crevasse-depth law, in

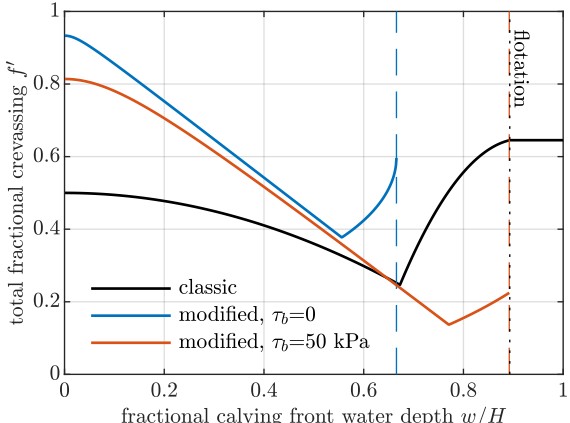

**Figure 3.** Illustrative example of the modified crevasse sizes, compared to the classic law (black), showing total fractional crevassing plotted against fractional ocean water depth at the calving front. The adjustable parameters take values $\rho_c = 1000$ kg m$^{-3}$, $H = 500$ m, and $\sigma_{max} =$ 150 kPa. The free slip case (blue) has $\tau_b = 0$ while the case with basal friction (red) has $\tau_b = 50$ kPa and $L = 500$ m. The dashed vertical lines indicate the water depth above which the modified crevasse sizes become undefined. The black dotted vertical line indicates the water depth beyond which the glacier is floating.

which the crevasse sizes are a function of the ratio $w/H$ only. We first consider an illustrative example with a fixed calving front ice thickness of $H = 500$ m, representing a relatively large tidewater glacier. We also take $\rho_c = 1000$ kg m$^{-3}$ (i.e., the basal crevasse is filled with freshwater), a tensile strength $\sigma_{max} = 150$ kPa and consider both free-slip ($\tau_b = 0$) and a case with

235 non-zero basal friction.

The total crevassed fraction $f' = (d'_s + d'_b)/H$ (sum of surface and basal crevassing) is shown in Fig. 3. In common with the classic crevasse-depth law, for dry calving cliffs or well-grounded glaciers, there are surface crevasses only (solution case 2 in section 2.4) and the crevassed fraction decreases as the water depth increases. However, the modified crevassed fractions in both the free slip and basal friction cases are significantly larger than in the classic law. From Eq. 18, and letting $\tilde{L} = L/H$

240 and $\tilde{\tau}_b = \tau_b/\rho_i gH$, basal crevasses are present (solution case 3 in section 2.4) when

$$\frac{w}{H} > \frac{\rho_i \tilde{\sigma}_{max}}{\rho_w - \rho_i} \left( 1 + \sqrt{1 + 2\frac{\rho_w - \rho_i}{\rho_w} \frac{\tilde{L}\tilde{\tau}_b}{\tilde{\sigma}_{max}^2}} \right), \tag{22}$$

unless this exceeds flotation, in which case there are no basal crevasses for any water depth. For the particular parameters we have chosen in Fig. 3, basal crevasses are present in the modified law for fractional water depths greater than $w/H = 0.56$ (free slip case) and $w/H = 0.77$ (basal friction case), observed as the sharp corner in the fractional crevassing. For larger water

245 depths, the crevassed fraction quickly increases as the basal crevasses get larger.

In the free slip case, above a critical fractional water depth of 0.65, the modified crevassed fraction becomes undefined (Fig. 3, solution case 4 in section 2.4), indicating that the horizontal force balance cannot be satisfied. In contrast, with basal friction, the crevasse sizes remain defined and the horizontal force balance can be satisfied for all water depths at which the

glacier is grounded. At flotation, however, the glacier loses its basal friction and the crevasse depths therefore become undefined
250  at flotation (Fig. 3).

## 3.2  Parameter sensitivity of modified crevasse sizes

The features observed in the illustrative example of Fig. 3 depend strongly on four parameters, these being the density of water in the basal crevasse, the ice tensile strength, the basal friction and the ice thickness. These sensitivities are illustrated in Fig. 4. Note that all of the statements discussing Fig. 4 should be seen in the context of a specific choice of parameters, rather than
255  applying to glaciers in general.

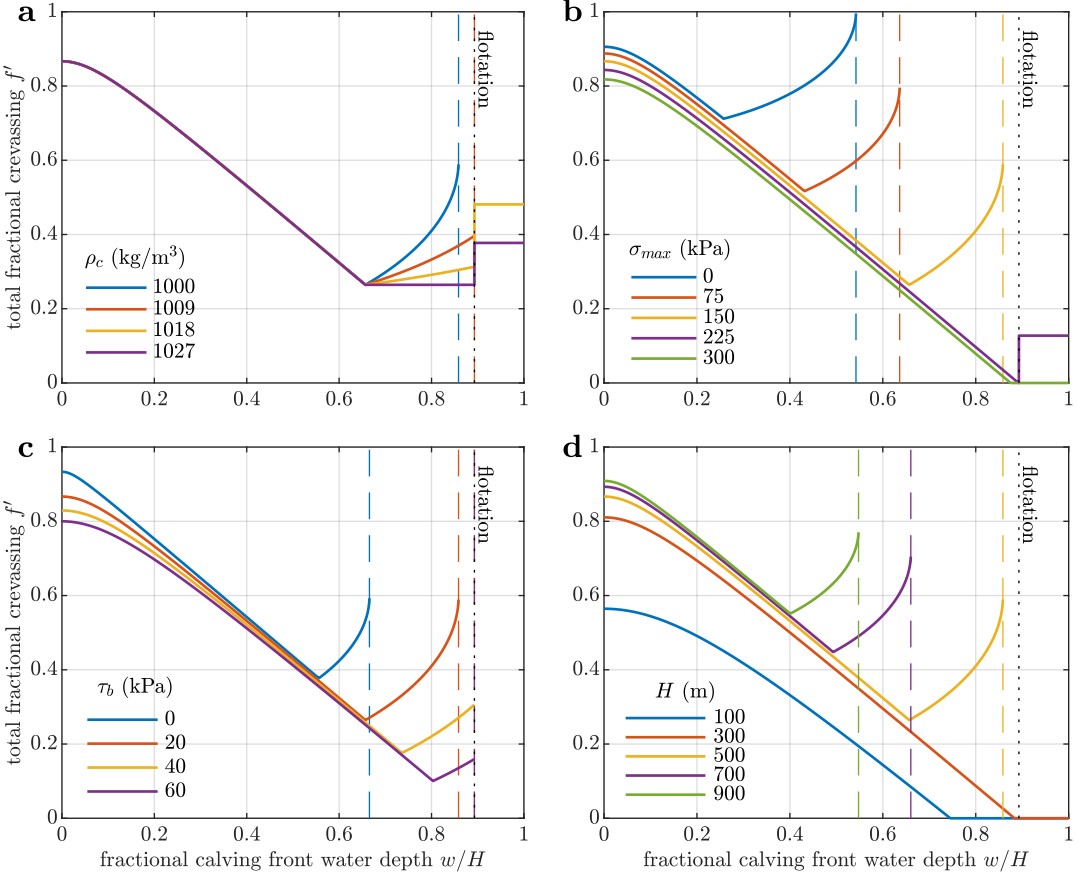

**Figure 4.** Parameter sensitivity of the modified crevasse depths, showing the influence of (a) crevasse water density, (b) ice tensile strength, (c) basal friction, and (d) ice thickness. Unless being varied, the crevasse water density is $\rho_c = 1000$ kg m$^{-3}$, the ice tensile strength is $\sigma_{max} = 150$ kPa, the ice thickness is $H = 500$ m, the basal friction is $\tau_b = 20$ kPa and the crevasse spacing $L$ is equal to the ice thickness. The coloured vertical dashed lines indicate the fractional water depth above which the modified crevasse sizes become undefined. The vertical dotted black lines show the water depth at which the front floats.

The sensitivity to basal crevasse water density is shown in Fig. 4a. Basal crevasse water density does not affect the result when only surface crevasses are present ($w/H < 0.65$ in Fig. 4a). Once basal crevasses are present, they grow much more quickly with water depth when the crevasse water density is smaller. Or equivalently, for a given water depth, basal crevasses are larger when the crevasse water density is smaller. This effect arises because when the crevasse water density is smaller, the water pressure in the crevasse decreases more slowly with height above the bed, hence the water pressure in the basal crevasse is higher. For the lower values of crevasse water density considered ($\rho_c = 1000$ and $1009 \, \text{kg m}^{-3}$), there are water depths beyond which the crevasse sizes are undefined; in the former case when the glacier is grounded and in the latter case at flotation. For crevasse water densities of 1018 and $1027 \, \text{kg m}^{-3}$, a stable solution for crevasse sizes exists for all fractional water depths, though there is a jump in crevassing at flotation when the glacier loses basal friction and the resistive stress thus increases suddenly. In our framework, therefore, the existence and impact of basal crevasses depends sensitively on the density of water in the basal crevasse.

Crevassing also depends strongly on the assumed ice tensile strength (Fig. 4b). For stronger ice (i.e., larger values of $\sigma_{max}$), the depth of surface crevasses is smaller and greater water depths are required to generate basal crevasses. For $\sigma_{max} = 300$ kPa, there are no basal crevasses for any water depth and no crevasses at all close to or at flotation. For $\sigma_{max} = 225$ kPa the loss of basal friction at flotation results in a jump in crevassed fraction. For $\sigma_{max} = 150$ kPa there are basal crevasses for $w/H > 0.66$ and crevasse depths become undefined when $w/H > 0.86$. The same is true at shallower water depths for $\sigma_{max} = 75$ and 0 kPa. The latter deserves particular attention: with zero tensile strength, the crevassed fraction equals 1 at the point where crevasse depths become undefined (Fig. 4b) – this special case occurs only with $\sigma_{max} = 0$ and $\tau_b > 0$ and is not the case with non-zero tensile strength.

The crevassed fraction is only weakly sensitive to basal friction for low water depths when there are only surface crevasses (Fig. 4c), because typical values of basal friction are small compared to the large resistive stress. At higher water depths, when the ice is close to flotation, the basal friction values shown become comparable to the resistive stress, so that basal friction affects both the water depth at which basal crevasses appear and the water depth at which calving occurs (Fig. 4c). Since basal friction disappears once the ice floats, the two highest basal friction values tested suggest calving would happen at flotation. Note that the results in Fig. 4c assume a crevasse spacing of $L = 500$ m and that the results are sensitive to this too, but since it is only the product $L\tau_b$ that appears in the modified crevasse depths, a plot showing sensitivity to crevasse spacing would be qualitatively similar to Fig. 4c.

Lastly, we can examine the sensitivity of crevassing to ice thickness (Fig. 4d). For sufficiently small ice thickness ($H = 100$ or 300 m in Fig. 4d, where $\sigma_{max} = 150$ kPa and $\rho_c = 1000 \, \text{kg m}^{-3}$), there are no basal crevasses present for any water depth. As ice thickness increases beyond this, basal crevasses form at smaller fractional water depths, and for sufficiently large water depths the crevasse sizes become undefined. For $H = 900$ m, there is no real solution for crevasse sizes beyond a fractional water depth of $w/H = 0.55$. The dependence of crevasse size on ice thickness in the modified crevasse-depth framework contrasts with the classic crevasse-depth law, in which there is no dependence on ice thickness. The difference arises from the introduction of a non-zero ice tensile strength in the modified framework.

## 3.3 Modified conditions for calving

### 3.3.1 Zero tensile strength, non-zero basal friction

In the crevasse-depth calving law, calving is conventionally taken to occur when the total fractional crevassing is equal to 1. For the modified crevasse depths close to calving fronts this can be achieved only when the ice has zero tensile strength ($\sigma_{max} = 0$) and there is non-zero basal friction (e.g., Fig. 4b). In this case, the condition for calving is obtained by solving for when the sum of Eqs. 19 & 20 is equal to one. This gives that calving occurs when the water depth exceeds a value

$$\frac{w_\tau}{H} = \sqrt{2\frac{\rho_i}{\rho_w - \rho_c}\frac{\rho_c}{\rho_w}\tilde{L}\tilde{\tau}_b}, \tag{23}$$

unless this exceeds flotation, in which case calving occurs at flotation because the basal friction disappears when the front floats. Put another way, if $w_\tau$ exceeds the water depth required for flotation, this means that the basal friction is sufficient to stabilise all grounded glaciers. But for any water depth above flotation, $\tilde{\tau}_b$ becomes 0, and without any tensile strength or basal friction, crevasse depths are undefined.

To understand what this means for calving at real glaciers, we have to decide whether the crevasse spacing $L$ and basal friction $\tau_b$ should vary with the ice thickness. While estimates of surface crevasse spacing are available (e.g., Enderlin and Bartholomaus, 2020), our crevasse spacing applies to pairs of surface and basal crevasses (Fig. 1), for which there are limited observational constraints. In what follows, we assume that the crevasse spacing and basal friction scale linearly with the ice thickness (so that $\tilde{L}$ and $\tilde{\tau}_b$ are constant) – this is both the simplest choice and is physically reasonable since larger glaciers generally calve larger icebergs (e.g., Åström et al., 2014) and stresses generally scale with ice thickness. This choice does not affect the end interpretation but does affect the parameter values that best fit the observations (Appendix B). Note that we view ice tensile strength $\sigma_{max}$ as a material parameter that is independent of ice thickness.

Under the choices $\tilde{L} = 1$ (so that the calved length is equal to the ice thickness) and $\tilde{\tau}_b = 0.013$ (so that for example, a glacier of 500 m thickness would have a basal friction of $\tau_b = 58$ kPa), the calving criteria under zero tensile strength is illustrated in Fig. 5a. This particular choice of basal friction is sufficient to make all grounded glaciers stable, so that according to Eq. 23 calving occurs at flotation for all ice thicknesses.

Fig. 5a also includes observed calving front ice thickness and water depth combinations from Ma et al. (2017). These observations were derived from IceBridge radar profiles (CReSIS, 2024) at 30 glaciers in Greenland, some at multiple points in time during 2006-2014. As noted in Ma et al. (2017), and similarly in Bassis and Walker (2012), these observations show glaciers that are almost all grounded but that tend to cluster near the line of flotation, particularly for large ice thicknesses. We also include six observed ice shelf calving fronts extracted from BedMachine for Greenland and Antarctica (Morlighem et al., 2017, 2020). The plotted values are the mean value across the full width of these calving fronts. The calving criterion provides a good bound on the observations from Ma et al. (2017), providing a physical reason for calving to occur at flotation. However, the assumption of zero tensile strength prohibits the existence of stable floating ice shelves, contrary to observations (Fig. 5a).

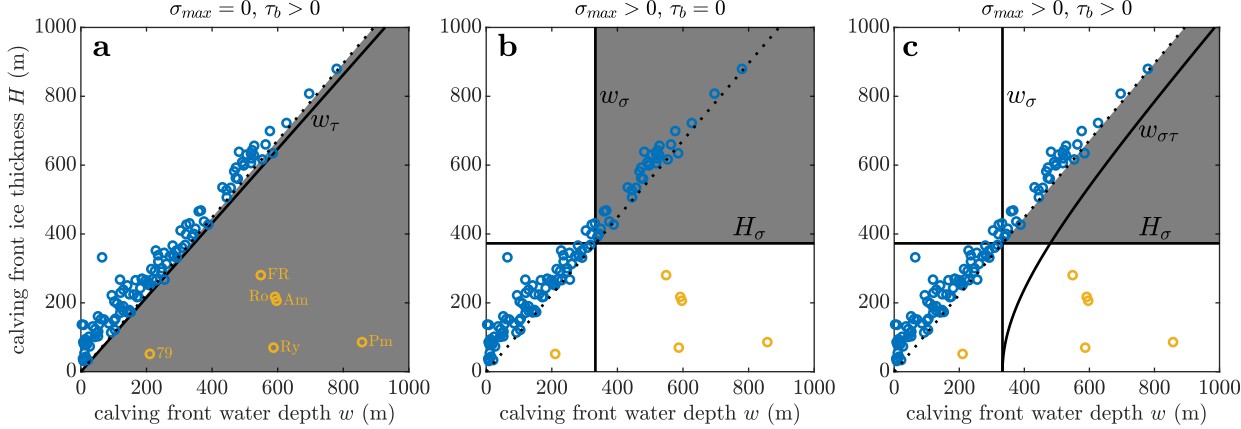

**Figure 5.** Illustration of the three possible modified calving criteria: (a) zero tensile strength, non-zero basal friction; (b) non-zero tensile strength, zero basal friction; (c) non-zero tensile strength, non-zero basal friction. Note that, when non-zero, basal friction is only applied to grounded glaciers. White regions are stable while grey shaded regions are combinations of ice thickness and water depth that would calve and should not exist according to the calving criteria set out in section 3.3. The critical water depths ($w_\tau$, $w_\sigma$, $w_{\sigma\tau}$) and the critical ice thickness ($H_\sigma$) are shown as black lines. Grounded calving fronts lie to the left of the black dotted line; floating calving fronts are to the right. The blue markers show observed calving front ice thickness and water depth values from Ma et al. (2017). The yellow markers are ice shelf front values extracted from BedMachine with abbreviations FR=Filchner-Ronne, Ro=Ross, Am=Amery (all Antarctica), and 79=79N, Ry=Ryder and Pm=Petermann (all Greenland). The parameters adopted (when non-zero) are $\rho_c = 1000$ $\text{kg}\,\text{m}^{-3}$, $\sigma_{max} = 150$ kPa, $\tilde{\tau}_b = 0.013$ and $\tilde{L} = 1$.

### 3.3.2 Non-zero tensile strength, zero basal friction

The other possibility for calving, seen throughout Figs. 3 & 4, is solution case 4 from section 2.4, in which the crevasse depths become undefined because it is not possible to balance the horizontal forces on the nascent calving block. In the case of free slip ($\tau_b = 0$), the crevasse sizes become undefined and calving occurs when the water depth exceeds a threshold value given by

$$w_\sigma = \left[ \frac{\rho_i \rho_c^2}{\rho_w(\rho_c - \rho_i)(\rho_w - \rho_c)} \right]^{1/2} \frac{\sigma_{max}}{\rho_i g}, \tag{24}$$

which is obtained by rearranging Eq. 21 after using Eqs. 6 & 8. The implication is that, for grounded glaciers with no basal friction, calving occurs when the absolute water depth exceeds $w_\sigma$. If the ice reaches flotation before it reaches $w_\sigma$, then calving never occurs by this criterion. For ice to reach flotation before reaching $w_\sigma$, the ice thickness must be less than

$$H_\sigma = \frac{\rho_w}{\rho_i} w_\sigma = \tilde{\rho} \frac{\sigma_{max}}{\rho_i g}, \tag{25}$$

where

$$\tilde{\rho} = \left[ \frac{\rho_w \rho_c^2}{\rho_i(\rho_c - \rho_i)(\rho_w - \rho_c)} \right]^{1/2}. \tag{26}$$

Provided that the ice tensile strength is not zero, this criterion, illustrated in Fig. 5b, is therefore an upper bound on the frontal water depth at grounded glaciers and an upper bound on the frontal ice thickness at ice shelves. The introduction of the non-zero tensile strength means that floating ice shelves are stable up to a critical frontal thickness given by Eq. 25, consistent with observations (Fig. 5b). However, this criteria suggests that all grounded or floating glaciers with frontal thickness greater than the critical value are unstable, which is contradicted by the observations (Fig. 5b).

### 3.3.3 Non-zero tensile strength, non-zero basal friction

This leads us to the final possibility, which is when both basal friction and ice tensile strength are non-zero. The water depth threshold for calving is modified from Eq. 24 to

$$w_{\sigma\tau} = w_\sigma \sqrt{1 + 2\left(1 - \frac{\rho_i}{\rho_c}\right)\frac{\tilde{\tau}_b\tilde{L}}{\tilde{\sigma}_{max}^2}}, \tag{27}$$

unless this exceeds flotation, in which case calving occurs at flotation when the ice is thicker than $H_\sigma$ (Eq. 25). The resulting calving criterion is visualised in Fig. 5c. The presence of the non-zero ice tensile strength ensures that all glaciers with a frontal thickness less than $H_\sigma$ are stable. For the chosen parameter values, $w_{\sigma\tau}$ exceeds flotation for all ice thicknesses, so that all grounded glaciers are stable and calving occurs at flotation when the frontal thickness exceeds $H_\sigma$. Put another way, the parameter space effectively splits into 3 regions (Fig. 5c). These are (i) $H < H_\sigma$, where glaciers are stabilised by the ice tensile strength; (ii) grounded glaciers with $H > H_\sigma$, which are stabilised by a combination of ice tensile strength and basal friction; and (iii) floating glaciers with $H > H_\sigma$, which are unstable as they experience no basal friction and the ice tensile strength is not sufficient to stabilise such glaciers. This final calving criterion is the only one in which all of the observed ice thickness and water depth combinations are stable (Fig. 5c), and is the criterion considered in the last part of the results.

### 3.4 Parameter sensitivity of calving criterion

Fig. 6 shows how the stable region according to the preferred criterion, lying to the left of and below the solid coloured lines, compares to observations, under varying basal crevasse water density (Fig. 6a) and ice tensile strength (Fig. 6b). Note that in comparison to Fig. 5c, we have omitted the shading in the unstable region so that we can show multiple parameter values. Larger basal crevasse water density or stronger ice increases the ice thickness threshold $H_\sigma$ (Eq. 25) above which calving can occur. Similarly, larger basal crevasse water density or stronger ice increases the water depth threshold $w_{\sigma\tau}$ (Eq. 27) above which calving occurs, but this has no influence on the stability envelope in Figs. 6a & b because this water depth threshold is greater than flotation and so calving instead occurs at flotation. In comparison to the observations, making the ice rather weak in the theory (say by adopting a value of $\sigma_{max} = 75$ kPa) would better explain why many glaciers with small ice thicknesses appear to be bounded by flotation, but this would then contradict the existence of the relatively thick Antarctic ice shelves (Fig. 6b).

Varying the magnitude of the basal friction (Fig. 6c) or the crevasse spacing (Fig. 6d) has no impact on the ice thickness threshold $H_\sigma$ above which calving occurs. Instead, reducing either the basal friction or the crevasse spacing decreases the

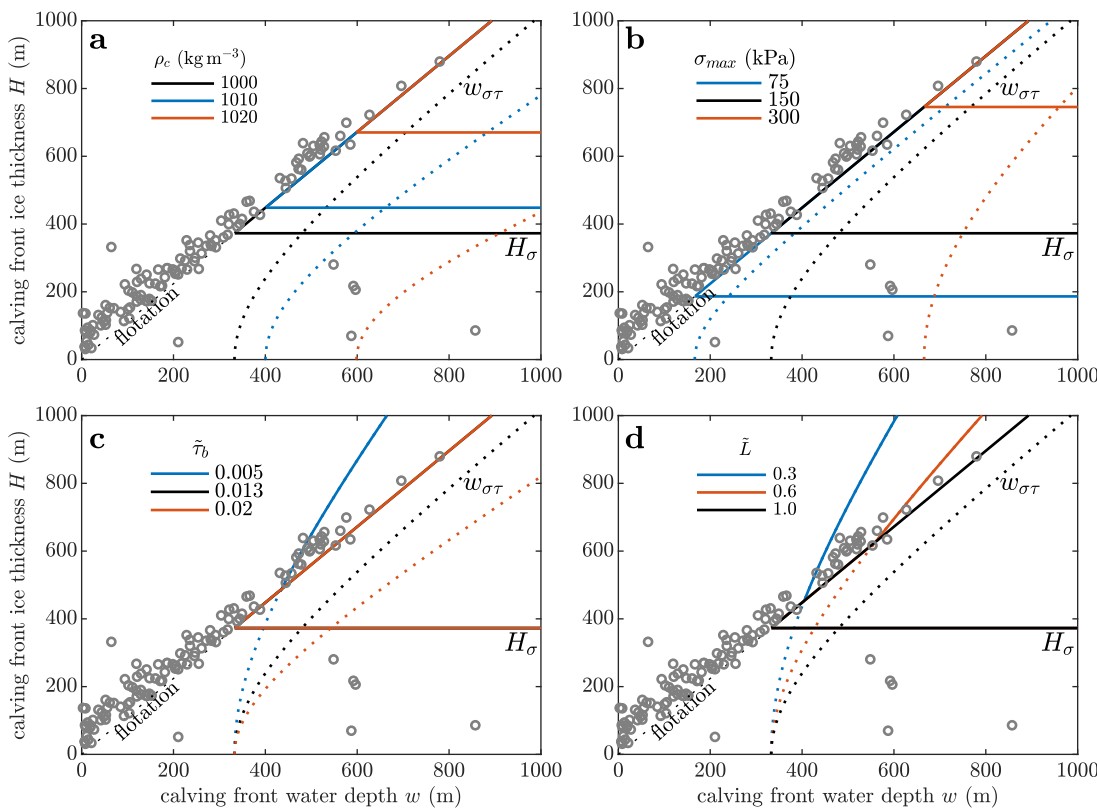

**Figure 6.** Version of Fig. 5c showing the sensitivity of the preferred calving criterion to (a) basal crevasse water density, (b) ice tensile strength, (c) basal friction and (d) crevasse spacing. When not being varied, the parameters take values $\rho_c = 1000\ \mathrm{kg\,m^{-3}}$, $\sigma_{max} = 150\ \mathrm{kPa}$, $\tilde{\tau}_b = 0.013$ and $\tilde{L} = 1$. For the indicated parameter values, frontal ice thickness and water depth combinations that lie left of and below the solid coloured lines are stable to calving (as was indicated by the unshaded region in Fig. 5c). The critical water depth $w_{\sigma\tau}$ is shown as dotted when it exceeds flotation (because then it is flotation that is providing the boundary for calving) and solid when it is less than flotation (because it then provides the boundary for calving). The grey markers show the same observed calving front ice thickness and water depth values as in Fig. 5. In (c), the solid black line lies beneath the solid orange line.

water depth threshold $w_{\sigma\tau}$ above which calving occurs. For sufficiently small basal friction or crevasse spacing, this water depth threshold can become shallower than that required to float the ice (Figs. 6c & d). In other words, the resistance to calving from basal drag on the nascent calving block becomes insufficient to prevent thick glaciers from calving. Thus, for example, the $\tilde{L} = 0.3$ and $0.6$ cases in Fig. 6d contradict the existence of large glaciers that are close to flotation, so that the theory becomes inconsistent with the observations at sufficiently small basal friction or crevasse spacing.

Comparing more generally the observed frontal geometries to the theoretical bounds, the main success of the theory is in providing an explanation, at least for thicker glaciers, for why many of the glaciers have flotation as a lower bound on their frontal ice thickness, while also allowing for the existence of ice shelves. The presence of the ice shelves for which we have

observations requires the tensile strength in the theory to be larger than around $\sigma_{max} = 120$ kPa (Fig. 6b, assuming $\rho_c = 1000$ kg m$^{-3}$). The stable existence of very thick, grounded glaciers requires the basal friction in the theory to be larger than around $\tilde{\tau}_b = 0.01$ (Fig. 6c, assuming $\tilde{L} = 1$), which corresponds to dimensional values of $\tau_b = 9$ kPa for a glacier of 100 m thickness or $\tau_b = 80$ kPa for a glacier of 900 m thickness.

## 4 Discussion

### 4.1 A modified crevasse-depth calving criterion

Motivated by the fact that the crevasse-depth approach is a natural and promising formulation for calving, but that in its classic form it predicts no calving, we have applied the approach of Buck (2023) to crevasses at grounded tidewater glaciers. Buck (2023) proposed a means of accounting for the feedback of crevasses on the stress field and allowed for the presence of freshwater in basal crevasses. We have adopted these concepts and incorporated the additional modifications of non-zero ice tensile strength and basal friction.

Each of these modifications is central to the outcome. Modifying only the density of water in basal crevasses does not give significantly larger crevasses (Fig. 2). Adding the feedback of crevasses on the stress field, while having zero tensile strength and no basal friction, gives no stable glaciers at all. The presence of non-zero tensile strength or non-zero basal friction gives the happy medium required for a calving law: that is, some calving front geometries are stable and some are unstable. Our preferred version of the theory has both non-zero ice tensile strength and non-zero basal friction. This gives a calving law under which observed calving fronts, both grounded and floating, are stable, while also providing a physical basis for why some glaciers calve at flotation.

Mathematically, calving in the preferred modified law appears in an unexpected way. Rather than the surface and basal crevasses stably and gradually fracturing the full thickness, as would happen in the classic law with water in the surface crevasses, it becomes impossible to satisfy the horizontal force balance for the nascent calving block so that the crevasse sizes become undefined. Physically, this might represent the situation where a grounded calving front is advancing into deeper water and the fractional crevassing is slowly increasing, until at a certain water depth the forces on the nascent calving block cannot be balanced, so the block accelerates relative to the glacier and the fracturing of the remaining ice thickness occurs very quickly.

Although the algebra is involved and there are a significant number of parameters in this modified formulation, the final modified calving criterion is actually rather simple (illustrated in the cartoon of Fig. 7). The modified formulation suggests that calving front ice thicknesses less than $H_\sigma = \tilde{\rho}\sigma_{max}/(\rho_i g)$ (Eq. 25) do not reach sufficiently high horizontal stresses to calve by this mechanism. With sufficiently large basal friction (see section 3.4), the framework becomes insensitive to the actual values of basal friction and crevasse spacing, and we are left with the simple criterion that fronts with ice thickness greater than $H_\sigma$ will calve at flotation, because basal friction is the factor stopping them from calving.

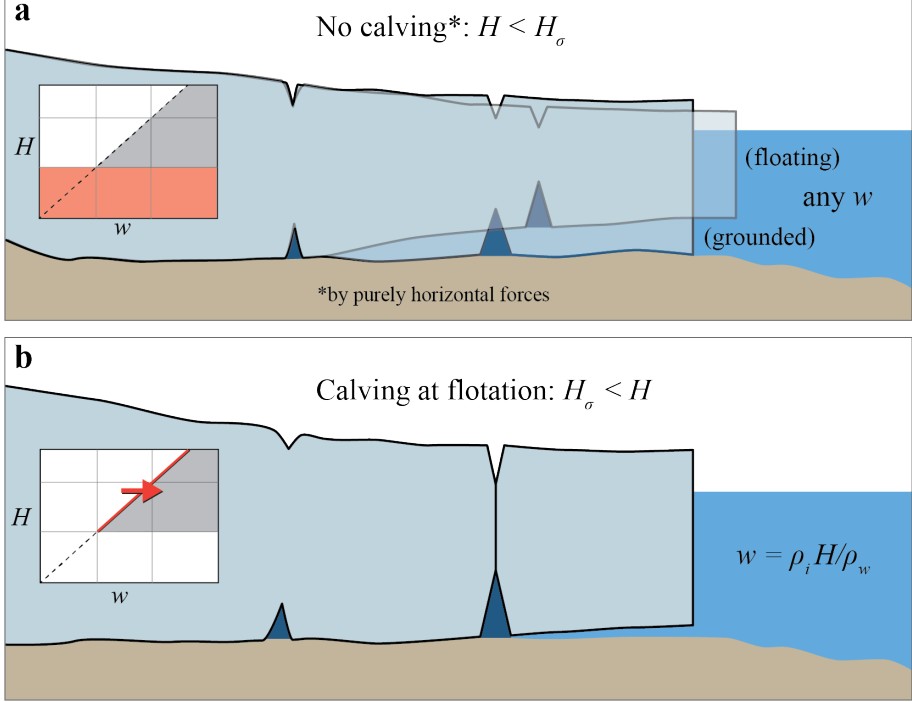

**Figure 7.** Cartoon of the two thickness regimes, illustrating the modified crevasse-depth calving criterion proposed here. The insets on the left show schematics of Fig. 5c, illustrating (a) the thickness regime for which no calving occurs (red rectangle) and (b) the transition from stable to unstable water depths for glaciers as they reach flotation (red arrow).

## 4.2 Interpretation in terms of calving styles

By the methodology that we have chosen, the calving style in this study is full-thickness and driven purely by horizontal forces. Our results suggest that this calving style occurs at flotation for calving front ice thicknesses greater than $H_\sigma$, but that it does not occur for calving front ice thicknesses less than $H_\sigma$, so that $H_\sigma$ appears as a thickness threshold above which this calving style appears (Figs. 5 & 6). If the basal crevasses are assumed to be filled with freshwater, and we take a value $\sigma_{max} = 150$ kPa for the effective tensile strength of ice (following results by Grinsted et al., 2024) then Eq. 25 gives $H_\sigma \approx 400$ m.

Goliber and Catania (2024) recently inferred the calving style of 10 glaciers in Greenland over recent decades, categorising into low-frequency, full-thickness events that they termed 'buoyant flexure', and high-frequency low-volume events that they termed 'serac failure'. We suggest that their buoyant flexure style, being full thickness and occurring close to flotation, is closely related to the full thickness calving that occurs in our framework for ice thicknesses above $H_\sigma$. In Goliber and Catania (2024), the three glaciers with the greatest frontal ice thickness (600–800 m) were assigned a dominant style of buoyant flexure. Three glaciers with small frontal ice thickness (100-300 m) were assigned a dominant style of serac failure. The remaining four glaciers had a mixed style, or a style that varied through time, and had intermediate frontal ice thickness (250–450 m). In particular, they showed that Sermeq Silardleq calved mostly by serac failure when its frontal ice thickness was approximately

300 m and, after retreating, calved mostly by buoyant flexure when its frontal ice thickness was approximately 450 m. From this and other studies (e.g., Fried et al., 2018; Bézu and Bartholomaus, 2024), there is therefore good observational evidence that calving style is influenced by ice thickness and that full-thickness calving occurs predominantly for thicker ice. Additionally, the thickness threshold above which full-thickness calving occurs in our theory, $H_\sigma \approx 400$ m, appears in good agreement between our theory and the observations.

The thickness threshold $H_\sigma$ is essentially the maximum ice thickness that unconfined, crevassed ice can support without horizontal stresses pulling it apart. At unconfined floating ice shelves, it should then provide an upper bound on the calving front thickness. The thickness threshold depends linearly on the assumed ice tensile strength $\sigma_{max}$, a parameter that observations have placed in the range of 90-320 kPa (Vaughan, 1993) and 110-200 kPa (Grinsted et al., 2024). The central value assumed in this study, $\sigma_{max} = 150$ kPa, lies within these estimates, and the resulting value of the thickness threshold, $H_\sigma \approx 400$ m, allows for the calving fronts of large Antarctic ice shelves such as the Ross, Filchner-Ronne and Amery to be stable (Fig. 5). Decreasing the tensile strength to $\sigma_{max} = 75$ kPa would contradict the existence of these Antarctic calving fronts (Fig. 6b) – assuming they are indeed unconfined – but does not contradict the existence of the Greenlandic floating calving fronts, which are much thinner.

For frontal ice thicknesses exceeding $H_\sigma$ (and for sufficiently large basal friction), our results suggest that calving should occur at flotation. As noted in Bassis and Walker (2012) and Ma et al. (2017), flotation does provide a good bound on the observed calving front ice thickness and water depth combinations of large Greenland tidewater glaciers. Bassis and Walker (2012), knowing that the classic crevasse-depth law did not give large enough crevasses to drive calving at flotation, provided a heuristic argument that this bound could be obtained analytically by assuming calving when ice at the base of the front reached a yield stress. Ma et al. (2017) considered an evolving glacier in a full-Stokes framework and found that glaciers with a free-slip basal boundary condition thinned to nearly flotation, at which point surface and basal crevasses intersected and calving occurred.

Our study provides a different route to the flotation bound that appears in observations. The bound in our study uses the analytical crevasse-depth framework, modified to account for the feedback between crevassing and the stress field, and arises because certain frontal ice thicknesses are stable in the presence of basal friction but unstable without. Therefore, when such a calving front reaches flotation it loses its basal friction, becomes unstable and calves. There is an important difference in our bound, relative to those before, however, which is that our bound applies only for ice thicknesses exceeding $H_\sigma$. This thickness scale is not present in previous studies such as Ma et al. (2017) and arises here due to the inclusion of a non-zero ice tensile strength (a similar scale does exist in linear elastic fracture mechanics approaches provided there is a non-zero fracture toughness).

Our results predict no calving at all for glaciers with frontal ice thicknesses less than $H_\sigma$ (Fig. 5), meaning that this formulation is incomplete as a calving law for Greenland's glaciers in general. But perhaps that is as it should be – this study has considered only horizontal forces, and so we could infer that calving at glaciers with smaller calving front ice thickness is not driven purely by horizontal forces. This is in agreement with studies of calving at smaller glaciers that suggest an important role for melt undercutting and serac failure (e.g., Luckman et al., 2015; Fried et al., 2018; How et al., 2019; Wagner et al.,

2019; Goliber and Catania, 2024; Bézu and Bartholomaus, 2024); processes in which rotational and vertical forces likely play an important role (Slater et al., 2021). Similarly, our results say nothing about the potential for ice cliff failure, which can provide an upper bound on ice thickness for grounded glaciers (Bassis and Walker, 2012). A more general formulation for calving will need to consider these processes too, and will therefore need to go beyond parameterisations or laws that consider only horizontal forces.

### 4.3 Role of basal friction

The presence of basal friction plays two roles in our results: (i) it reduces the resistive stress that drives calving; (ii) it disappears at flotation, leading to the behaviour of calving at flotation. While basal friction is the unique process that fulfils (ii), other processes can contribute to (i), such as lateral drag transmitted from fjord walls or buttressing by ice mélange and sea ice. These process have the potential to modify the thresholds and required parameter values presented in this study.

The inclusion of basal friction within the calving law itself, as opposed to simply as a boundary condition on ice flow, is a subtle issue. There are two ways that our results could be used within an ice sheet model. An ice sheet model could apply Eqs. 15-20, substituting in the value of $R_{xx}$ that is calculated by the model, and look for regions where the total fractional crevassing reaches 1 or is undefined. Or, following our analysis that uses the boundary value of $R_{xx}$ (Eq. 8), an ice sheet model could instead apply the final calving criterion (Fig. 7) as a condition on the frontal ice thickness. In the former, basal friction enters the stress balance that the model uses to estimate $R_{xx}$, but it does not enter the calving criterion. In the latter, basal friction enters the calving criterion, which also introduces a typical crevasse spacing $L$. This notion of a discrete crevasse spacing would not be present in the first approach to implementing our results in an ice sheet model. Since we have argued that the presence of crevasses results in stress concentrations that in turn increase the size of crevasses, it is natural that calving is most likely to occur at crevasses that are advected into the near-terminus region from upstream. From this perspective, it is natural that crevasse spacing would appear in the calving criterion, from which it follows that basal friction on the nascent calving block should be accounted for.

### 4.4 Limitations

Having discussed where we feel this revised crevasse-depth law matches observations and has important implications, it seems befitting to also consider its weaknesses and the assumptions that were necessary to reach the results.

Firstly, the treatment of stress is highly idealised. This includes the flowline nature of the analysis, and the assumptions of hydrostasy and depth-invariant deviatoric stress close to the calving front and in the presence of crevassing. We feel that these assumptions are justified within the context of an analytical study, and because many of the models that need a better treatment of calving are depth-integrated and so do not resolve variability with depth. Full-Stokes models suggest that these variations in stress close to the front can play a role in calving, and the question remains whether we inescapably need these complex models or whether revised analytical approaches, like that presented here, can be sufficient.

Basal drag on the nascent calving block plays a key role in our results but its treatment is very simplified with both basal friction and crevasse spacing assumed to scale with ice thickness. This is a pragmatic choice for a simplified theoretical study

and we have discussed the sensitivity to this choice in section 3.4 and Appendix B. Commonly used sliding laws relate basal friction to the ice velocity and to effective pressure, both of which are likely to depend on the ice thickness and the water depth (e.g., Schoof and Hewitt, 2013), thus driving variation in the basal friction across the parameter space we have considered in, for example, Fig. 5. A development of our framework could seek to apply a sliding law in place of our simplified treatment of basal friction.

Similarly, our treatment of the materials involved is simplified, as we have adopted uniform densities of ice and water, uniform ice tensile strength and uniform basal friction, all quantities that may vary in reality. Our results are also quite sensitive to these parameters, as was shown in Fig. 4. The density of the water filling the basal crevasse is a particular unknown – for a well-grounded glacier it seems reasonable that basal crevasses would be filled with freshwater, but for glaciers closer to flotation it is increasingly recognised that seawater may reach inland of the grounding line (Wilson et al., 2020; Kim et al., 2024). What helps, however, is that despite all of the parameters involved in the analysis, the implications for calving essentially boil down to the thickness threshold $H_\sigma$ and whether basal friction is sufficient to stabilise thick glaciers (Figs. 6c & d). If confident in the analysis that leads to this, one could potentially tune the thickness directly rather than the underlying physical parameters.

Lastly, real glaciers may have non-vertical calving fronts. This does not directly affect the horizontal force balance that is central to this study (i.e., Eq. 8 still holds for a non-vertical calving front), but submarine melt-induced undercutting of the calving front does remove a region of ice-bed contact and will therefore affect the ice thickness thresholds that we have suggested characterise calving.

## 5 Conclusion

The crevasse-depth calving law is a promising candidate for parameterising calving at marine-terminating glaciers, but struggles to predict sufficiently deep crevasses to trigger calving. Following Buck (2023), we modified the crevasse depth law to account for the stress concentration under crevassing, and allowed for a variable density of water in basal crevasses. We made additional modifications to allow for non-zero ice tensile strength and basal friction. We provide revised estimates for surface and basal crevasse size (Eqs. 15-20) and show that the revised estimates can give full-thickness crevassing without appealing to water in surface crevasses.

The revised crevasse-depth criterion suggests that calving driven purely by horizontal forces (for that is what the crevasse-depth law accounts for) splits into two regimes depending on the ice thickness (Fig. 7). For calving front ice thickness less than a threshold value $H_\sigma$ (Eq. 25), the horizontal forces alone are not large enough to drive calving. For calving front ice thickness greater than $H_\sigma$, calving occurs when the ice reaches flotation (though this second regime can be modified slightly if basal friction is weak). Our best estimates for the physical parameters involved – and in particular assuming an ice tensile strength of 150 kPa and that basal crevasses are filled with freshwater – give a value for $H_\sigma$ of roughly 400 m. Thus, the revised crevasse-depth law provides an explanation for observations showing that grounded glaciers with frontal ice thickness exceeding 400 m have a dominant calving style of infrequent full-depth events while grounded glaciers with smaller thicknesses

calve more frequent, serac-type icebergs. It also suggests that unconfined floating ice shelf fronts should not exceed roughly 400 m thickness.

We propose that this revision of the crevasse-depth criterion is a step closer to a better understanding of the calving process, but it is incomplete as a calving law because it provides no reason for glaciers with frontal ice thickness less than 400 m to calve. Rather than seeing this as a limitation, we feel this is appropriate because submarine melting likely plays an important role in calving at such glaciers, and melt undercutting induces significant rotational and vertical imbalances. A more unified treatment of calving will need to treat consistently the horizontal forces that are encapsulated by the crevasse-depth law with the rotational and vertical imbalances that it currently ignores.

## Appendix A: Relation to previous floating ice shelf analysis

To see how our analysis and expressions connect with those provided for ice shelves by Buck (2023) and Coffey et al. (2024), start from our Eq. 19, but set $H_{ab} = 0$ to enforce flotation to get

$$\frac{d'_b}{H} = \frac{\rho_i}{\rho_c}\left[1 - \frac{\rho_c\tilde{\sigma}_{max}}{\rho_c - \rho_i} - \sqrt{1 - \frac{2\rho_c\tilde{R}_{xx}}{\rho_c - \rho_i} + \left(\frac{\rho_c\tilde{\sigma}_{max}}{\rho_c - \rho_i}\right)^2}\right]. \tag{A1}$$

If we further note that the resistive stress at the front of a floating ice shelf (Eq. 8) is given by

$$R^0_{xx} = \frac{1}{2}\rho_i g H\left(1 - \frac{\rho_i}{\rho_w}\right), \tag{A2}$$

then we can write the basal crevasse height as

$$\frac{d'_b}{H} = \frac{\rho_i}{\rho_c}\left[1 - \frac{\rho_c\tilde{\sigma}_{max}}{\rho_c - \rho_i} - \sqrt{1 - \frac{\rho_c}{\rho_w}\frac{\rho_w - \rho_i}{\rho_c - \rho_i}\frac{R_{xx}}{R^0_{xx}} + \left(\frac{\rho_c\tilde{\sigma}_{max}}{\rho_c - \rho_i}\right)^2}\right]. \tag{A3}$$

The corresponding surface crevasse depth (again setting $H_{ab} = 0$) is given from Eq. 20 as

$$\frac{d'_s}{H} = \frac{\rho_c - \rho_i}{\rho_i}\frac{d'_b}{H}. \tag{A4}$$

As described in the three cases in section 2.4, these hold provided that $R_{xx} > \sigma_{xx}$; if not, then there are no surface or basal crevasses. Eqs. A3 & A4 are the the same solution as Buck (2023) but with non-zero ice tensile strength. Thus, if we assume zero ice tensile strength ($\sigma_{max} = 0$) then we get equivalent expressions to those in Buck (2023), and if we further take the basal crevasse to be filled with seawater ($\rho_c = \rho_w$) then the basal crevasse height simplifies to

$$\frac{d'_b}{H} = \frac{\rho_i}{\rho_w}\left(1 - \sqrt{1 - \frac{R_{xx}}{R^0_{xx}}}\right), \tag{A5}$$

which is the (isothermal) solution in Coffey et al. (2024).

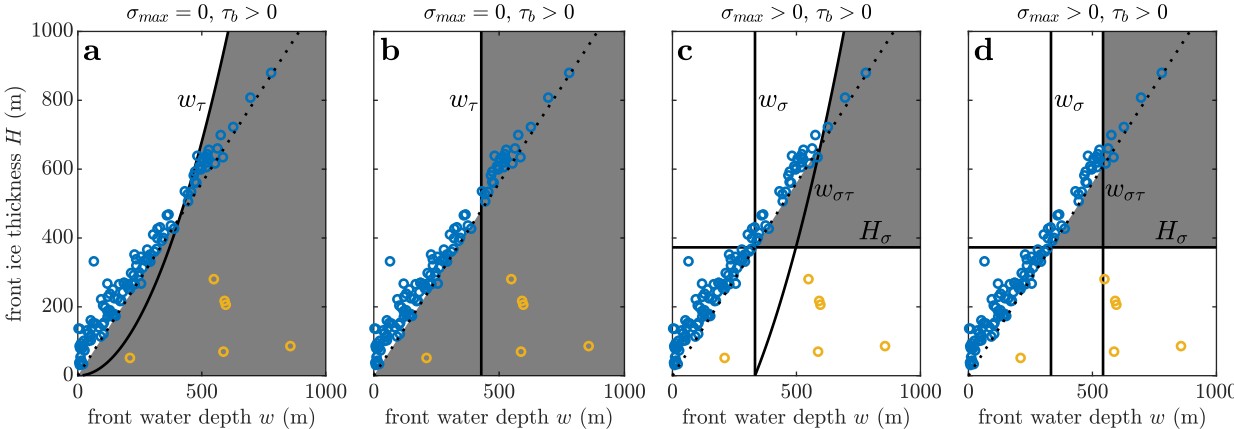

**Figure B1.** An illustration of the impact on the calving criterion of making different assumptions of how basal friction and crevasse spacing vary with ice thickness. (a): Version of Fig. 5a (i.e., assuming zero ice tensile strength but non-zero basal friction) but taking basal friction to be constant at 50 kPa across all ice thicknesses. (b): as for (a), but additionally taking crevasse spacing to be constant at 500 m across all ice thicknesses. (c): Version of Fig. 5c (i.e., assuming non-zero ice tensile strength and non-zero basal friction) but taking basal friction to be constant at 50 kPa across all ice thicknesses. (d): as for (c), but additionally taking crevasse spacing to be constant at 500 m across all ice thicknesses.

## Appendix B: Impact of choice of scaling for basal friction and crevasse spacing

In the main results we chose to scale the basal friction and the crevasse spacing with ice thickness, so that both are proportionally larger at thicker grounded glaciers. If instead we assume that basal friction is constant regardless of ice thickness, with $\tau_b = 50$ kPa, then Fig. 5a becomes Fig. B1a and Fig. 5c becomes Fig. B1c. The impact on the calving criterion (i.e., the grey shaded region) is that the thickest glaciers would calve at a shallower water depth than flotation. Larger values of basal friction ($\tau_b = 100$ kPa in Fig. B1a and $\tau_b = 80$ kPa in Fig. B1c) would make the water depth thresholds $w_\tau$ and $w_{\sigma\tau}$ deeper so that all of the observations lie in the white region. If both basal friction and crevasse spacing are assumed to be constant regardless of ice thickness, with $L = 500$ m, then the water depth thresholds become independent of ice thickness (Fig. B1b and d). In this case, either the basal friction or crevasse spacing can be increased so that the calving criterion matches the observations, but we do not consider this case to be physically realistic since we expect that the size of the calved block should depend on the ice thickness. A more complete treatment of basal friction could apply a full sliding law to estimate basal friction across the ice thickness-water depth parameter space.

*Code and data availability.* As a largely theoretical paper, this paper used no substantial code. The only data used are the ice thickness/frontal water depth data for Greenland marine-terminating glaciers, available as a supplement to Ma et al. (2017).

*Author contributions.* DAS conceived the study and the analysis. DAS and TJWW undertook the analysis and wrote the paper.

*Competing interests.* No competing interests are present.

*Acknowledgements.* DAS acknowledges support from NERC Independent Research Fellowship NE/T011920/1. TJWW was supported by the NSF Office of Polar Programs through grants # 2148544 and # 2338057.

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
