# Peer review of "Calving from horizontal forces in a revised crevasse-depth framework"

_EGUsphere, 2024_

## Author Comment (AC1)

We thank the editor and both reviewers for their comments and their time spent considering our manuscript. We are pleased that the manuscript was well-received overall. Below, we offer initial responses to the reviewer and editor comments, in the hope that we are asked to submit a revision in which we can give full responses and revise the manuscript accordingly. The reviewer/editor comments are in black, ours are in blue italics.

From these reviews, we see three significant issues:

**1. Structure and flow of the paper**

It is clear from general comment 1 of reviewer 1, much of the general comment of reviewer 2, and the initial comments of the editor that the flow of the manuscript needs revision. We tried a narrative style where the reader is led through the manuscript in the same manner that we discovered the results, but this has proven confusing because it mixes methods and results, and some results that come later in the paper "overwrite" results from earlier in the paper. If asked to submit a revision, we will revise the structure so that the new calving formulation in its most general form is introduced up front and then we zoom in to the specific results in the later sections.

**2. Interpretation of "undefined crevasse depths" and when calving occurs**

Reviewer 1 twice raises a concern about the interpretation of undefined crevasse depths and the definition of calving. Mathematically, these undefined crevasse depths arise when the term in the square roots of Eqs. 15 & 16 (which define the crevasse depths) becomes negative, so that there is no real solution. Physically, this indicates that the horizontal forces on the nascent calving block (Fig. 1) cannot be balanced. Or, in other words, the resistive stress that opens crevasses exceeds the tensile strength of the ice, for any configuration of crevasse depths. We feel that it is natural to call this situation "calving" – the forces that resist calving are unable to balance the forces that promote calving.

As pointed out by reviewer 1, this definition differs from Buck (2023) and Coffey et al. (2024), both of whom define calving as occurring when the total fractional crevassing, f, is equal to 1. We agree that f=1 would signify calving and if that arose in our results, we would interpret that as calving. But, perhaps more importantly, a smooth approach to f=1 does not often arise in our results for two reasons. First, we are considering significantly higher resistive stresses than Buck or Coffey. Their papers consider buttressed floating ice shelves (with a limiting case of a freely floating ice shelf), whereas we consider grounded tidewater glaciers. The resistive stress at the front of a grounded tidewater glacier is higher than at a buttressed or freely floating ice shelf, which leads to larger (or even undefined) crevasse depths. Second, the variable that scales the resistive stress in our results is the fractional calving front water depth w/H (i.e., how close to flotation the glacier is). This water depth also controls the water pressure in the basal crevasse, so the resistive stress and the water pressure in the basal crevasse are both responding to the calving front water depth. This differs from Buck and Coffey because their resistive stress is varied independently from the water pressure in the basal crevasse.

Summing up, the undefined crevasse depths in our results look unfamiliar in the context of the results of Buck and Coffey, but they arise due to the different physical situation (grounded glaciers here versus ice shelves in their work). If invited to submit a revision, we will add a new section covering these points.

**3. Novelty relative to Buck (2023)**

Reviewer 2 raises the issue of the novelty of our results relative to Buck (2023). At a superficial level, the difference is that we apply the revised crevasse depth framework to grounded tidewater glaciers, whereas Buck (2023) applied it to buttressed floating ice shelves. Thus, our results pertain to the vast majority of Greenland's outlet glaciers and other grounded tidewater glaciers around the globe, whereas the results of Buck pertain to the floating ice shelves around Antarctica.

But, maybe more importantly, the dynamics that emerge from applying the framework to grounded tidewater glaciers are very different from floating ice shelves, with key differences being:

- The resistive stresses close to the front of grounded tidewater glaciers are higher than at floating ice shelves, meaning that for much of the grounded glacier parameter space there are no stable crevasse depths (i.e., horizontal forces cannot be balanced).
- We introduce a non-zero ice tensile strength, which was not considered in Buck (2023). As well as being an important factor to introduce in general (because the fracture of any material should depend on its intrinsic strength), the tensile strength results in a threshold ice thickness ($H_0$ = 400 m approx., Eq. 20) that shows promise in explaining the separation between different calving styles of grounded glaciers (L346-364).
- We consider the role of basal friction in suppressing calving. A key property of basal friction is that it disappears when the ice reaches flotation, and thus the presence of basal friction in the new calving framework provides a potential explanation for why many glaciers appear to calve at or close to flotation, as shown by the data from Ma et al. (2019) that is plotted on Figs. 7 and 9.

Thus, although the initial theoretical basis is the same as Buck (2023), we present a different physical application that is in a different part of the "parameter space", has quite different dynamics, and we introduce new ideas that show promise in explaining key overarching calving dynamics observed at grounded glaciers. For these reasons we feel this study is a significant contribution in its own right. These points will be made clearer in a revision, particularly in the writing of the revised methodology (see also response 1 above), through a better overall flow of the paper, and by specifically addressing differences in the discussion.

**Reviewer 1**

General comments

Following Buck (2023), The authors modified the crevasse depth law with a simple method to account for the stress concentration under crevassing, a variable density of water in basal crevasses, and non-zero ice tensile strength. The framework has the potential to understanding differing calving styles and a better parameterization of calving in numerical models. However, there are major concerns that need to be resolved before considering publication.

1. The structure of the analytical model can be confusing. The heart of the analytical model – the horizontal force balance (Eqn11) – involves many hidden assumptions that the authors prove to be invalid in sections afterwards. For instance, Eqn11

assumes no basal friction (proved to be inaccurate in Sec.3.5), static condition (proved to be wrong when there is no real solution to Eqn15,16, line 335 "the forces on the nascent calving block cannot be balanced, so the block accelerates relative to the glacier…"). To improve readability of the paper, it might be better to start with the most general form of force balance by taking basal friction and acceleration into account in Eqn.11.

Please see our response to the first significant concern above.

2. I'm confused by the authors' definition of calving: when there is no real solution to surface/basal crevasse depth (Eqn15,16), which, to me, implies invalid assumptions underlying current analytical model. For instance, in Fig4, an increase in rho_c will give real solutions to Eqn15,16. Does this mean rho_c=1000 kg/m^3 is just not physical? In line 85, the authors state that "it is assumed there is an open hydraulic connection from the bed below the glacier to the calving front", does this contradict with the later assumption of rho_c=1000kg/m^3? If we incorporate an inertia term in Eqn. 11, will it be guaranteed to have real solutions instead?

Please see our response to the second significant concern above.

As a side note, calving was previously defined as "total fractional crevassing $f = 1$" in literature (Buck, 2023, Coffey et al., 2023 Theoretical stability of ice shelf basal crevasses with a vertical temperature profile), which seems more intuitive and physical to me. I'm curious whether there are plausible ways to modify the analytical model (i.e., relax some assumptions), to avoid the "undefined crevasse fraction" issue.

Please see our response to the second significant concern above.

1. Section 3.2, shall the tensile strength be nondimensionalized by rho_i*g*H_i? Does it vary with ice thickness? The authors might want to justify why in many figures tensile strength is fixed while ice thickness is varying across a wide range? (Fig.5, 6, 7, 8, 9)

Thank you for raising this point. The ice tensile strength could be non-dimensionalised for mathematical convenience, but in general it should be independent of ice thickness because it is a material parameter – i.e., it is a physical property of ice – which is why it is fixed (except in Fig. 4b where we consider sensitivity to the ice tensile strength). Ice thickness is varied across a wide range because we wish to consider the implications of our formulation at the full range of Greenland's tidewater glaciers. If asked to submit a revision we will make these motivations clear.

2. Figure 7, 9: The original paper for the observational data also has a stability phase diagram of terminus configuration (Figure 3 in Ma et al., 2017), and it seems to explain the data better. It is reasonable that the authors (no shear failure) have different predictions than Ma et al., 2017 (including shear failure). However, in Ma et al., 2017, ice tensile strength is 0 (Figure 1), which is claimed to be invalid in this manuscript instead? Why is there a contradiction?

This is an interesting point. We don't feel there is a contradiction between our results and those of Ma et al. (2017), rather that we make different assumptions about the ice tensile strength. Ma et al. (2017) assume that ice tensile strength is 0, so that tensile failure occurs when the (maximum principal) stress is greater than 0, whereas we assume a non-zero tensile

strength and that tensile failure occurs only when the stress exceeds this tensile strength. In Ma et al. (2017), calving occurs close to flotation at all ice thicknesses (their Fig, 3), whereas in our results calving occurs close to flotation only above a certain ice thickness (Fig. 9). If Ma et al. (2017) had assumed a non-zero tensile strength, we think it is likely that their simulations with smaller ice thicknesses would not undergo calving because the stresses would not exceed the tensile strength, bringing their results into line with ours.

You make the point that their results seem to explain the data better, and that is a fair point. However we think there is value in presenting our results for two reasons: (i) ice has a non-zero tensile strength, and it is natural that this should be a parameter taken into account in calving laws, and (ii) observations indicate that the dominant calving style of glaciers changes with ice thickness (see L352-364 of our manuscript); our results provide a possible physical explanation for this, whereas calving style is uniform across all ice thickness values in Ma et al. (2017). Again, we can make these points clear in a revised manuscript.

Specific comments

1. Figure 5: A question comes to mind that "why happens for thick termini that are at floatation? (commonly observed across Greenland)" Does the figure imply termini thicker than 300m can never be at floatation stably?

That is indeed what is implied by this figure (in fact, the threshold is close to 400 m – see Fig. 6). However, this assumes a crevasse water density of 1000 kg/m3 and a tensile strength of 150 kPa, sensitivities which are explored in Fig. 7. We also state in L277 that this "does not look very promising as a calving law", precisely for the reason you state that we do observe thicker glaciers at flotation in Greenland. This then is the primary motivation for including basal friction in the section that follows (section 3.5), so that on Fig. 9 you can see that the formulation does allow for stable, thicker glaciers at flotation. Perhaps again what is confusing here is the way we have laid out the narrative in the manuscript, and as stated above we will improve this in a revision.

2. Line 245, "when we do assume zero tensile strength, the total fractional crevassing is either 1 or undefined for all possible calving fronts, hence this is not a useful calving law". The sentence could be rephrased. This conclusion is drawn based on assumptions made in this paper but might not be general?

Yes – this is drawn based on the framework in this paper. We will aim to clarify this statement by rewording along the lines "when considering the force balance at the calving front in the present framework with zero tensile strength, the total fractional crevassing is either 1 or undefined for all possible calving fronts and therefore does not provide a non-trivial calving law".

3. Line 285: "if we use Eq.22 in the modified crevasse sizes Eqs. 15& 16 then for grounded glaciers, calving occurs…" Eqs. 15 & 16 is derived by assuming no basal friction in Eqn. 11, right? Is it equivalent between: 1) what the authors did here; and 2) including tau_b in Eqn. 11 -> rederive Eqn. 12 -> combining 12,13,14 -> expression for ds'/H, db'/H by considering basal friction. Does 2) seem easier to understand?

Yes – your suggestion would be equivalent – thank you for the suggestion. As discussed above, we will restructure the derivation of these results and begin with the most general version, which will hopefully clear up this source of confusion.

4. Figure 9: the observations have different crevasse spacing, L, right? If the calving criterion is very sensitive to L, maybe the authors should include the information of L for the observation data as well, before comparing with the analytical model?

Thank you for raising this point. The calving criterion is moderately sensitive to the assumed value of L (compare the blue and purple lines on Fig. 9, for which all parameters are the same except that the blue line has L=500 m and the purple has L=250 m). Unfortunately, observational data for crevasse spacing are limited, especially for basal crevasses, and so we do not have the necessary information to estimate a crevasse spacing for every glacier on Fig. 9. Within the scope of this present study, we feel it is sufficient to show the sensitivity to crevasse spacing and we can raise this point in a revised discussion.

**Reviewer 2**

GENERAL COMMENT:

The manuscript "Calving from horizontal forces in a revised crevasse-depth framework" by Slater and Wagner modifies the 'classic' crevasse-depth calving law by introducing feedback with the background stress field and the water density within basal crevasses. This work builds on recent analytical formulations to improve the 'classic' crevasse-depth law in a largely theoretical exercise. The manuscript is of immediate interest of folks in the cryosphere community and is appropriate for the readership of the Cryosphere journal.

Thank you for your interest and time spent on this review.

While the manuscript provides a thorough introduction to calving laws, I found it quite difficult to follow. The structure poses some challenges, particularly in the introduction. Although the discussion of calving laws is detailed, it took me several readings to fully understand it. For instance, it wasn't immediately clear that Nye (1955) is considered the 'classic' crevasse-depth law (is that correct?). There have been many modifications since Nye's original stress-balance formulation, and the authors do a good job to list them. However, later in the methodology sections (i.e. 2.2), the manuscript includes a detailed formulation of this approach, but it's unclear where these equations originate from—Nye? Weertman? Furthermore, in line 111, the 'line of argument' from Buck (2023) is introduced, which raises the question: are we still discussing the original formulation? While I'm not suggesting the formulas are incorrect, these sections are quite confusing, and the definition of the 'classic' formulation remains ambiguous. Following this, Buck (2023)'s modifications are presented in Section 2.3, but the integration of equations (1–2) with the classic formulation (equations 3–8) is unclear. After reading these sections, I got completely lost on the novelty of this paper leaving me with this question: What is the main difference with Buck 2023?

Please see our response to the first and third significant concerns above.

Finally, in the results section (i.e. 3.5), the equations are further edited with the inclusion of basal friction. I appreciated this part, and I think it is an important aspect of the manuscript given the ongoing discussion in the community surrounding the importance of sliding laws.

However, this section mixes methods and results, which had to bring me back to the initial equations, further slowing down the reading process. Wouldn't it be perhaps more efficient to introduce the novel formulation at the beginning of the methodology section and move the classic laws in the appendix? I know this would be a major edit for this manuscript and ultimately it is the authors' personal choice, but it is a scenario that is perhaps worth considering.

Please see our response to the first significant concern above.

To be clear, I am not suggesting that this work lacks novelty, and I commend the authors for the detailed mathematical analysis presented—it is evident that this required significant time and effort. However, the structure in which the methods and results are presented makes the manuscript occasionally slow and difficult to follow. These issues should be addressed to improve clarity and accessibility before publication.

Please see our response to the first significant concern above.

MINOR COMMENTS:

Line 20: There is also a large body of literature which analyze the how individual rifts and the modulation of their infill (ice melange) can lead to calving, whilst in complete absence of the factors listed here (e.g. Borstad 2017 GRL and Larour 2021 PNAS). This aspect is important to mention here.

Thank you - we can add this point to a revised manuscript.

Line 192: I am not sure what is meant with 'an example of the modified crevasse sizes'.

At this point we wish to show a specific example of how the modified crevasse formulation that we have introduced differs from the classic (Nye) crevasse formulation. We can make this clearer in a revision, and the issue may also be helped by an overall restructuring of methods and results as proposed in response to the first significant concern above.

Line 316: Unfortunately, this sentence at this point of the manuscript leaves me questioning the novelty again. What is the main difference with Buck 2023?

Please see our response to the third significant concern above.

Line 320: See my general comment above.

Please see our response to the first significant concern above.

**Specific Editor comments:**

The revised formulation introduces several tuneable parameters, many of which are difficult to quantify—particularly the variable water density in basal crevasses, which plays a key role in determining calving sensitivity. Without validation through real-world observations or detailed model simulations, it remains unclear whether the resulting formulation may be overly dependent on parameter tuning.

This is a fair concern. Our first response would be that the introduction of more realistic physical processes into the calving formulation does indeed bring more physical parameters, but this isn't necessarily a bad thing provided that the formulation is more physically accurate. We do also address the problem of parameter sensitivity in L416-422 where we note that despite the many input parameters, the only things that really matter in the end (i.e., for a calving law) are the two thickness scales H0 and H1. Ultimately, we completely agree that candidate calving laws need validation and that is something we wish to do in future work.

Another concern is that the observational data used for comparison are clustered in a narrow envelope near the flotation criterion (e.g., Fig. 9a). The authors predict that under conditions of high ice thickness and low water depth, the ice front should remain stable, yet no corresponding observations are provided to support this claim.

Yes – the formulation that we have put forward does indeed suggest that high ice thickness/low water depth combinations are stable, but this arises because we have not considered shear failure. Bassis & Walker (2012) and Ma et al. (2017) both showed that shear failure prevents high ice thickness/low water depth combinations from being stable, which explains why there are no observations of glaciers in that part of the parameter space on Fig. 9. We addressed this issue briefly in L342-344 but in a revision we can make this clearer and more prominent. We have not considered shear failure in this manuscript because it would add significant additional complexity to do so, and we feel that the manuscript is already relatively complex and makes a significant contribution even without shear failure. We hope the editor understands this motivation.

---

## Author Response (AR1)

We thank the editor and both reviewers for their comments and their time spent considering our manuscript. We are pleased that the manuscript was well-received overall. Below, we offer our responses to the reviewer and editor comments and explain the corresponding changes we have made to the manuscript. The reviewer/editor comments are in black, ours are in red. Where section, figure or line numbers are given these refer to the clean version of the revised manuscript (the version with changes marked is hard to read within the methods and results section because we have revised these significantly).

We first offer an overall response to the main issues raised.

**1. Structure and flow of the paper**

It is clear from general comment 1 of reviewer 1, much of the general comment of reviewer 2, and the initial comments of the editor that the flow of the manuscript needed revision. We tried a narrative style where the reader is led through the manuscript in the same manner that we discovered the results, but we agree that this proved confusing because it mixed methods and results, and some results that came later in the paper "overwrote" results from earlier in the paper. We have now:

- Comprehensively restructured the methods (section 2) and results (section 3) following the suggestions of the reviewers. The equations and solutions are stated in their most general form up front and before any results are presented.
- We have included basal friction from the start (Fig. 1 and Eq. 8 onwards) and throughout the results (Fig. 3 onwards).
- We have described and discussed the potential for it to be impossible to satisfy horizontal force balance at the first place where we solve the equations (L195).

**2. Interpretation of "undefined crevasse depths" and when calving occurs**

Reviewer 1 twice raises a concern about the interpretation of undefined crevasse depths and the definition of calving. Mathematically, these undefined crevasse depths arise when the term in the square root of Eq. 19 becomes negative, so that there is no real solution. Physically, this indicates that the horizontal forces on the nascent calving block (Fig. 1) cannot be balanced. Or, in other words, the resistive stress that opens crevasses exceeds the tensile strength of the ice, for any configuration of crevasse depths. We feel that it is natural to call this situation "calving" – the forces that resist calving are unable to balance the forces that promote calving.

As pointed out by reviewer 1, this definition differs from Buck (2023) and Coffey et al. (2024), both of whom define calving as occurring when the total fractional crevassing, f, is equal to 1. We agree that f=1 should signify calving, and in the revised manuscript we now include results where we do have f=1 (Fig. 4b, blue; section 3.3.1). We overlooked this possibility in the original manuscript, however for the reasons outlined in section 3.3.1 we do not think it is the most physical of the possibilities for calving and it has not changed the manuscript overall.

We still maintain that the undefined crevasse depths should be interpreted as calving and feel that these did not arise in the work of Buck or Coffey for two reasons. First, we are considering significantly higher resistive stresses than Buck or Coffey. Their papers consider buttressed floating ice shelves (with a limiting case of a freely floating ice shelf), whereas we

consider grounded tidewater glaciers. The resistive stress at the front of a grounded tidewater glacier is higher than at a buttressed or freely floating ice shelf, which leads to larger (or even undefined) crevasse depths – and it is natural that if total fractional crevassing reaches 1 at the front of an unconfined floating ice shelf (Buck 2023, Coffey 2024), then crevasse depths are at risk of becoming undefined when the resistive stress is significantly higher. Second, the variable that scales the resistive stress in our results is the fractional calving front water depth w/H (i.e., how close to flotation the glacier is). This water depth also controls the water pressure in the basal crevasse, so the resistive stress and the water pressure in the basal crevasse are both responding to the calving front water depth. This differs from Buck and Coffey because their resistive stress is varied independently from the water pressure in the basal crevasse. We have now:

- Included a possible solution where the total fractional crevassing reaches 1 (section 3.3.1) and Fig. 5a.
- Described in more detail why we think the undefined crevasse depths should be interpreted as calving and included this formally in the solution of the equations (L195-203).
- Described the differences between Buck, Coffey and our results in L210-220 and derived the connection between our mathematical expressions and theirs in Appendix A.

**3. Novelty relative to Buck (2023)**

Reviewer 2 raises the issue of the novelty of our results relative to Buck (2023). At a superficial level, the difference is that we apply the revised crevasse depth framework to grounded tidewater glaciers (with floating glaciers as an edge case), whereas Buck (2023) applied it to buttressed floating ice shelves. Thus, our results pertain to, for example, Greenland's outlet glaciers, whereas the results of Buck pertain to the floating ice shelves around Antarctica. But, maybe more importantly, the dynamics that emerge from applying the framework to grounded tidewater glaciers are very different from floating ice shelves, with key differences being: (i) the resistive stresses close to the front of grounded tidewater glaciers are higher than at floating ice shelves, meaning that for much of the grounded glacier parameter space there are no stable crevasse depths (i.e., horizontal forces cannot be balanced); (ii) we introduce a non-zero ice tensile strength, which was not considered in Buck (2023). As well as being an important factor to introduce in general (because the fracture of any material should depend on its intrinsic strength), the tensile strength results in a threshold ice thickness (Eq. 25, 400 m approx.) that shows promise in explaining the separation between different calving styles of grounded glaciers (L400-412); (iii) we consider the role of basal friction in suppressing calving. A key property of basal friction is that it disappears when the ice reaches flotation, and thus the presence of basal friction in the new calving framework provides a potential explanation for why many glaciers appear to calve at or close to flotation, as shown by the data from Ma et al. (2019) that is plotted on Fig. 5.

Thus, although the initial theoretical basis is the same as Buck (2023), we present a different physical application that is in a different part of the "parameter space", has quite different dynamics, we introduce new ideas that show promise in explaining key overarching calving dynamics observed at grounded glaciers, and we offer observational support (Fig. 5). For these reasons we feel this study is a significant contribution. In terms of changes to the manuscript, we have:

- Modified the abstract to clarify the advance and difference to Buck (2023) – L5
- Made clearer that non-zero ice tensile strength and basal friction are factors that were not considered in Buck (2023) – L77, 373, 489.
- Explicitly discussed differences in the theoretical framework between this study, Buck (2023) and Coffey et al. (2024) – L208-218.
- Shown how our results map onto Buck (2023) when you (i) assume flotation and (ii) assume zero ice tensile strength – Appendix A.

**Reviewer 1**

General comments

Following Buck (2023), The authors modified the crevasse depth law with a simple method to account for the stress concentration under crevassing, a variable density of water in basal crevasses, and non-zero ice tensile strength. The framework has the potential to understanding differing calving styles and a better parameterization of calving in numerical models. However, there are major concerns that need to be resolved before considering publication.

1. The structure of the analytical model can be confusing. The heart of the analytical model – the horizontal force balance (Eqn11) – involves many hidden assumptions that the authors prove to be invalid in sections afterwards. For instance, Eqn11 assumes no basal friction (proved to be inaccurate in Sec.3.5), static condition (proved to be wrong when there is no real solution to Eqn15,16, line 335 "the forces on the nascent calving block cannot be balanced, so the block accelerates relative to the glacier…"). To improve readability of the paper, it might be better to start with the most general form of force balance by taking basal friction and acceleration into account in Eqn.11.

Please see our response to the first significant concern above – in summary, we have comprehensively restructured much of the methods (section 2) and results (section 3) to start with the most general formulation and avoid 'overwriting' results from earlier in the paper. In particular, basal friction is now included at the earliest opportunity (Fig. 1 and Eq. 8) and integrated throughout the results section. The potential for it to be impossible to satisfy the static condition is noted as a formal part of the crevasse size solution (case 4 of the 4 solution cases presented in L179-198), and we immediately discuss our interpretation of this in L197-203.

We considered adding an explicit acceleration term to the horizontal balance, and appreciate the suggestion, but felt that this would complicate the maths without adding physical insight. Ultimately, we would find that a non-zero acceleration is necessary to solve the equations in solution case 4 (L195), and then we would have the same physical interpretation as outlined in L197-203. Since we have worked hard to try to present the maths as simply as possible, we wish to avoid further complication and hope the reviewer understands this perspective.

2. I'm confused by the authors' definition of calving: when there is no real solution to surface/basal crevasse depth (Eqn15,16), which, to me, implies invalid assumptions underlying current analytical model. For instance, in Fig4, an increase in rho_c will give real solutions to Eqn15,16. Does this mean rho_c=1000 kg/m^3 is just not physical? In line 85, the authors state that "it is assumed there is an open hydraulic

connection from the bed below the glacier to the calving front", does this contradict with the later assumption of rho_c=1000kg/m^3? If we incorporate an inertia term in Eqn. 11, will it be guaranteed to have real solutions instead?

Please see our overall responses 1 and 2, and our previous response, for changes made to the paper to address these points. Addressing explicitly the questions posed here, we don't feel that undefined crevasse depths invalidate the analytical model. Rather, our interpretation is that undefined crevasse depths means that it is not possible to satisfy the static equations with any combination of crevasse depths. This means there must be an acceleration of the nascent calving block, which we would interpret as calving. This is now outlined in L195-203.

In Fig. 4 we agree there are cases where increasing rhoc will give real solutions, but we don't feel this is problematic or that this means rhoc=1000 kg/m3 is unphysical. Higher rhoc means a reduction in overall basal crevasse water pressure (L256) and therefore a reduction in the overall force opening the basal crevasse. The reduction can be sufficient that it becomes possible to satisfy the static balance, leading to real solutions. Our interpretation would then be that rhoc=1000 kg/m3 will rapidly lead to calving, whereas a higher value of rhoc does not result in calving.

By "open hydraulic connection" we mean there is a communication of pressure (so that the pressure at the bottom of the crevasse is the same as the pressure at the bottom of the ocean), rather than that the densities of crevasse and ocean water have to be the same. This has been clarified in L90.

As a side note, calving was previously defined as "total fractional crevassing f = 1" in literature (Buck, 2023, Coffey et al., 2023 Theoretical stability of ice shelf basal crevasses with a vertical temperature profile), which seems more intuitive and physical to me. I'm curious whether there are plausible ways to modify the analytical model (i.e., relax some assumptions), to avoid the "undefined crevasse fraction" issue.

Hopefully our responses so far now cover this point. Note that in revising the paper we discovered a solution we had overlooked that does have f=1 (section 3.3.1), but we do not favour this overall as a modified calving criterion because it does not allow for the existence of floating ice shelves and does not explain the threshold in calving styles discussed in section 4.2.

1.  Section 3.2, shall the tensile strength be nondimensionalized by rho_i*g*H_i? Does it vary with ice thickness? The authors might want to justify why in many figures tensile strength is fixed while ice thickness is varying across a wide range? (Fig.5, 6, 7, 8, 9)

Thank you for raising this point. The ice tensile strength is occasionally non-dimensionalised in the paper for mathematical convenience, but in general it should be independent of ice thickness because it is a material parameter – i.e., it is a physical property of ice – which is why it is fixed in many figures. This has been noted in L301. Ice thickness is varied across a wide range because we wish to consider the implications of our formulation at the full range of Greenland's tidewater glaciers.

2.  Figure 7, 9: The original paper for the observational data also has a stability phase diagram of terminus configuration (Figure 3 in Ma et al., 2017), and it seems to explain the data better. It is reasonable that the authors (no shear failure) have different predictions than Ma et al., 2017 (including shear failure). However, in Ma et

al., 2017, ice tensile strength is 0 (Figure 1), which is claimed to be invalid in this manuscript instead? Why is there a contradiction?

This is an interesting point. We don't feel there is a contradiction between our results and those of Ma et al. (2017), rather that we make different assumptions about the ice tensile strength. Ma et al. (2017) assume that ice tensile strength is 0, so that tensile failure occurs when the (maximum principal) stress is greater than 0, whereas we assume a non-zero tensile strength and that tensile failure occurs only when the stress exceeds this tensile strength. In Ma et al. (2017), calving occurs close to flotation at all ice thicknesses (their Fig. 3), whereas in our results calving occurs close to flotation only above a certain ice thickness (Fig. 5b-c and Fig. 6). If Ma et al. (2017) had assumed a non-zero tensile strength, we think it is likely that their simulations with smaller ice thicknesses would not undergo calving because the stresses would not exceed the tensile strength, bringing their results into line with ours. We have made a minor change at L434 to reflect this as we feel that paragraph (L429-436) covers this issue, but we can make further changes if deemed necessary.

You could ask why then we prefer having non-zero ice tensile strength? This is because of the need to explain the existence of unconfined ice shelves (which otherwise cannot exist – section 3.3.1 and Fig. 5a) and because the ice tensile strength leads to the thickness threshold which appears to map onto a threshold in calving behaviour (section 4.2).

Specific comments

1. Figure 5: A question comes to mind that "why happens for thick termini that are at floatation? (commonly observed across Greenland)" Does the figure imply termini thicker than 300m can never be at floatation stably?

This is now Fig. 4d. These results are intended to illustrate the sensitivities present in the framework, rather than being results that apply to glaciers in general. This has been clarified in L253. The revised results section 3.3, and particularly Fig. 5, cover the stability of glaciers in general.

2. Line 245, "when we do assume zero tensile strength, the total fractional crevassing is either 1 or undefined for all possible calving fronts, hence this is not a useful calving law". The sentence could be rephrased. This conclusion is drawn based on assumptions made in this paper but might not be general?

This sentence has been removed in the overall restructuring of the methods and results. In particular, section 3.3.1 of the results now explicitly considers the case of zero tensile strength.

3. Line 285: "if we use Eq.22 in the modified crevasse sizes Eqs. 15& 16 then for grounded glaciers, calving occurs…" Eqs. 15 & 16 is derived by assuming no basal friction in Eqn. 11, right? Is it equivalent between: 1) what the authors did here; and 2) including tau_b in Eqn. 11 -> rederive Eqn. 12 -> combining 12,13,14 -> expression for ds'/H, db'/H by considering basal friction. Does 2) seem easier to understand?

Yes – your suggestion would be equivalent – thank you for the suggestion. As stated above, we have now restructured the methods following your suggestions.

4. Figure 9: the observations have different crevasse spacing, L, right? If the calving criterion is very sensitive to L, maybe the authors should include the information of L for the observation data as well, before comparing with the analytical model?

Figure 9 is now Figure 6b. Thank you for raising this point. Unfortunately, observational data for crevasse spacing are limited, especially for basal crevasses, and so we do not have the necessary information to estimate a crevasse spacing for every glacier on this figure. Within the scope of this present study, we feel it is sufficient to consider the sensitivity to the assumed crevasse spacing, and this is now done in L279-281, L300-304 and Appendix B.

**Reviewer 2**

GENERAL COMMENT:

The manuscript "Calving from horizontal forces in a revised crevasse-depth framework" by Slater and Wagner modifies the 'classic' crevasse-depth calving law by introducing feedback with the background stress field and the water density within basal crevasses. This work builds on recent analytical formulations to improve the 'classic' crevasse-depth law in a largely theoretical exercise. The manuscript is of immediate interest of folks in the cryosphere community and is appropriate for the readership of the Cryosphere journal.

Thank you for your interest and time spent on this review.

While the manuscript provides a thorough introduction to calving laws, I found it quite difficult to follow. The structure poses some challenges, particularly in the introduction. Although the discussion of calving laws is detailed, it took me several readings to fully understand it. For instance, it wasn't immediately clear that Nye (1955) is considered the 'classic' crevasse-depth law (is that correct?). There have been many modifications since Nye's original stress-balance formulation, and the authors do a good job to list them. However, later in the methodology sections (i.e. 2.2), the manuscript includes a detailed formulation of this approach, but it's unclear where these equations originate from—Nye? Weertman? Furthermore, in line 111, the 'line of argument' from Buck (2023) is introduced, which raises the question: are we still discussing the original formulation? While I'm not suggesting the formulas are incorrect, these sections are quite confusing, and the definition of the 'classic' formulation remains ambiguous. Following this, Buck (2023)'s modifications are presented in Section 2.3, but the integration of equations (1–2) with the classic formulation (equations 3–8) is unclear. After reading these sections, I got completely lost on the novelty of this paper leaving me with this question: What is the main difference with Buck 2023?

Finally, in the results section (i.e. 3.5), the equations are further edited with the inclusion of basal friction. I appreciated this part, and I think it is an important aspect of the manuscript given the ongoing discussion in the community surrounding the importance of sliding laws. However, this section mixes methods and results, which had to bring me back to the initial equations, further slowing down the reading process. Wouldn't it be perhaps more efficient to introduce the novel formulation at the beginning of the methodology section and move the classic laws in the appendix? I know this would be a major edit for this manuscript and ultimately it is the authors' personal choice, but it is a scenario that is perhaps worth considering.

To be clear, I am not suggesting that this work lacks novelty, and I commend the authors for the detailed mathematical analysis presented—it is evident that this required significant time and effort. However, the structure in which the methods and results are presented makes the manuscript occasionally slow and difficult to follow. These issues should be addressed to improve clarity and accessibility before publication.

Please see our overall responses 1 and 3 at the top of this document – we have comprehensively restructured the methods and results sections following your comments here.

We now see that the introduction of the classic law could be confusing, particularly because did blend a bit the classic law with Buck (2023) by allowing for a variable density of water in the basal crevasse. Our motivation for doing so was that we wished to demonstrate how that modification alone has limited effect on crevasse depths (Fig. 2). We have rephrased this section (2.2) to clarify our definition of the classic law, to make clear what references the equations come from, and to avoid the confusion of mentioning Buck (2023) at this stage. We have also added a clarifying statement on the terminology 'classic' at the end of section 2.1 (L106).

Note that we have not moved the "classic" law to an appendix because we have found in discussing this work with others (and even with researchers who are familiar with the subject) that this is useful background. We also think it is worth recapping how we can use the calving front boundary condition to switch from thinking about crevasse depth as a function of resistive stress to thinking about crevasse depth as a function of frontal water depth. We have now made this an explicit section (section 2.3) to clarify this point. Essentially, we feel that including a recap of the more familiar "classic" law should help more readers to understand what follows, and hope the reviewer understands this motivation.

MINOR COMMENTS:

Line 20: There is also a large body of literature which analyze the how individual rifts and the modulation of their infill (ice melange) can lead to calving, whilst in complete absence of the factors listed here (e.g. Borstad 2017 GRL and Larour 2021 PNAS). This aspect is important to mention here.

Thank you – this has been added (L22).

Line 192: I am not sure what is meant with 'an example of the modified crevasse sizes'.

At this point we wish to show a specific example of how the modified crevasse formulation that we have introduced differs from the classic crevasse formulation. We hope that the overall restructuring of the methodology makes this clearer.

Line 316: Unfortunately, this sentence at this point of the manuscript leaves me questioning the novelty again. What is the main difference with Buck 2023?

Line 320: See my general comment above.

Please see our responses above. Note that for these specific lines (now L373 onwards), we have clarified in the following sentences what we have added.

**Specific Editor comments:**

The revised formulation introduces several tuneable parameters, many of which are difficult to quantify—particularly the variable water density in basal crevasses, which plays a key role in determining calving sensitivity. Without validation through real-world observations or detailed model simulations, it remains unclear whether the resulting formulation may be overly dependent on parameter tuning.

This is a fair concern. Our first response would be that the introduction of more realistic physical processes into the calving formulation does indeed bring more physical parameters, but this isn't necessarily a bad thing provided that the formulation is more physically accurate. However, we do also address the problem of parameter sensitivity in L391-396 where we note that despite the many input parameters, it is quite possible that the only parameter that the final calving criterion relies on is the thickness threshold $H_{\sigma}$. Ultimately, we completely agree that candidate calving laws need validation and that is something we wish to do in future work.

Another concern is that the observational data used for comparison are clustered in a narrow envelope near the flotation criterion (e.g., Fig. 9a). The authors predict that under conditions of high ice thickness and low water depth, the ice front should remain stable, yet no corresponding observations are provided to support this claim.

Yes – the formulation that we have put forward does indeed suggest that high ice thickness/low water depth combinations are stable, but this arises because we have not considered shear failure. Bassis & Walker (2012) and Ma et al. (2017) both showed that shear failure prevents high ice thickness/low water depth combinations from being stable, which explains why there are no observations of glaciers in that part of the parameter space on Fig. 5 (formerly Fig. 9). We have not considered shear failure in this manuscript because it would add significant additional complexity to do so, and we feel that the manuscript is already relatively complex and makes a significant contribution even without shear failure. We have however acknowledged this point at L443.

---

## Author Response (AR2)

*Thank you to the editor and both reviewers for once again taking the time to comment on our manuscript. Below we describe how we have addressed these comments. Line numbers refer to the version of the manuscript with changes marked on.*

**Editor**

Figure 5: This figure is generally clear. However, as noted by one of the reviewers, the assumption L = H is rather strong, especially for Greenlandic glaciers. In Fig. 6, you show $\tilde{L} = 0.5$, but it would be beneficial to the readers to illustrate how the results change for different values of L/H , such as 0.3 and 0.6. Similarly, since $\sigma_{\max}$ is not an observable parameter, further insight into its choice of this value would be beneficial. I therefore suggest expanding Fig. 6 into four sub plots, each varying a different parameter across at least three representative values.

*Thanks for this suggestion – we have expanded Fig. 6 into four subplots, covering the range of values requested, and modified the associated text in section 3.4 accordingly. In addition to the figure, we have added discussion on the value of $\sigma_{\max}$ in the paragraph beginning on L425 and have added to the manuscript on the subject of L=H – see response to reviewer comment below.*

Figure 6a: It is particularly interesting that even with $\sigma_{\max} = 75k$, the three Greenlandic glaciers remain within the calving regime. Please consider expanding your discussion to provide more insight into the potential physical meaning of $\sigma_{\max}$, including the rationale behind the selected values.

*We've added additional justification and discussion of the choice of $\sigma_{\max}$ to the discussion section (paragraph beginning L425). We're not sure we follow, in this particular context, why the case of the Greenlandic glaciers with tensile strength of 75 kPa is particularly interesting – the tensile strength sets the critical ice thickness that unconfined, crevassed ice can support without horizontal stresses pulling it apart. The Greenlandic floating fronts are relatively thin, perhaps due to higher submarine melt rates than in Antarctica. In this sense, the Greenlandic fronts are a bit less useful/interesting in constraining the critical ice thickness than the Antarctic fronts. Of course, our study only provides this upper bound on unconfined ice thickness and is not able to say what drives calving for ice shelves that are within the stable envelope. This likely involves processes such as melt hydrofracturing or frontal bending, but these are beyond the present study. We hope the additional discussion suffices in this case.*

Abstract and Conclusions: You mention $H_0$ is about 400m, but this value depends on the assumptions $\sigma_{\max} = 150k$ and $\rho = 1000$. While I am comfortable with skipping these assumptions in the abstract, please ensure they are explicitly stated in the conclusions for clarity and transparency.

*We've added this explicit statement to the conclusions (L518). We also fixed an inconsistency where $H_0$ should be denoted $H_{\sigma}$ according to Eq. 25.*

Line 117 & 178: Please use proper math notation: $R_{xx} / (\rho_i g H)$.

*Fixed as suggested.*

Line 179: Do you mean \sigma_{\max}?

*Yes – thanks for catching this.*

**Reviewer**

I'm confused by Fig5. The observational data points: blue dots are for grounded glacier, which should use tau_b>0 to compare with; yellow dots are floating ice shelves, which should use tau_b=0 to compare with. Shall the authors come up with two separate stability diagrams for a) sigma_max>0, tau_b>0 to compare with blue dots; and b) sigma_max>0, tau_b=0 to compare with yellow dots? For instance, in Fig.5b, only yellow dots should be presented; in Fig.5c, only blue dots should be presented. Did I misunderstand anything?

*Apologies – we can see the potential for confusion here. In Fig. 5, basal friction is only applied when \tau_b>0 AND when the front is grounded (i.e., in panels a and c, left of the flotation line). Basal friction is never applied to floating fronts. We have modified the caption of Fig. 5 to make this clearer.*

Can authors explain a bit more why they choose crevasse length = ice thickness (L=H)? In Appendix B it seems the results are sensitive to the choice of L. From TRI field data, the crevasse spacing seems to vary from 300m~600m, for thickness ~1km glaciers: https://tos.org/oceanography/article/an-intensive-observation-of-calving-at-helheim-glacier-east-greenland https://tc.copernicus.org/articles/12/1387/2018/tc-12-1387-2018.pdf

*Thank you for these sources and this suggestion. Following the suggestion from the editor, we've added a new subplot (Fig. 6d) showing specifically the sensitivity to the choice of L/H, including the cases L/H=0.3 and 0.6 that would be indicated by the sources you provided. The impact on the results is that for smaller crevasse spacing, the resistive to calving from basal friction is not enough to stop the thickest glaciers from being unstable (Fig. 6d). Since these glaciers do stably exist, we have favoured L/H=1 in most of the results. Note that our crevasse spacing applies to pairs of surface and basal crevasses (Fig. 1), whereas the observations linked by the reviewer are only able to see surface crevasses, which may be spaced more closely than basal crevasses. More broadly, our treatment of basal friction is very simplified and using a proper sliding law would be more realistic, so we don't wish to place too much weight on the precise values here, rather we want to make a sensible choice and then show the sensitivity to this choice. These thoughts have been added to the manuscript at L304, L368 and L488.*